# Structures of the T cell potassium channel Kv1.3 with immunoglobulin modulators

Purushotham Selvakumar[1,8], Ana I. Fernández-Mariño[2,8], Nandish Khanra[1], Changhao He[1], Alice J. Paquette[3], Bing Wang[3], Ruiqi Huang[4,5], Vaughn V. Smider[4,5,6], William J. Rice [3,7], Kenton J. Swartz [2] & Joel R. Meyerson [1✉]

The Kv1.3 potassium channel is expressed abundantly on activated T cells and mediates the cellular immune response. This role has made the channel a target for therapeutic immunomodulation to block its activity and suppress T cell activation. Here, we report structures of human Kv1.3 alone, with a nanobody inhibitor, and with an antibody-toxin fusion blocker. Rather than block the channel directly, four copies of the nanobody bind the tetramer's voltage sensing domains and the pore domain to induce an inactive pore conformation. In contrast, the antibody-toxin fusion docks its toxin domain at the extracellular mouth of the channel to insert a critical lysine into the pore. The lysine stabilizes an active conformation of the pore yet blocks ion permeation. This study visualizes Kv1.3 pore dynamics, defines two distinct mechanisms to suppress Kv1.3 channel activity with exogenous inhibitors, and provides a framework to aid development of emerging T cell immunotherapies.

[1] Department of Physiology and Biophysics, Weill Cornell Medical College, New York, NY, USA. [2] Molecular Physiology and Biophysics Section, Porter Neuroscience Research Center, National Institute of Neurological Diseases and Stroke, National Institutes of Health, Bethesda, MD, USA. [3] Cryo-Electron Microscopy Core, New York University School of Medicine, New York, NY, USA. [4] Applied Biomedical Science Institute, San Diego, CA, USA. [5] Minotaur Therapeutics, San Diego, CA, USA. [6] Department of Molecular Medicine, The Scripps Research Institute, La Jolla, CA, USA. [7] Department of Cell Biology, New York University School of Medicine, New York, NY, USA. [8]These authors contributed equally: Purushotham Selvakumar, Ana I. Fernández-Mariño. ✉email: jrm2008@med.cornell.edu

The Kv1.3 channel is a voltage-activated potassium (Kv) channel that is abundantly expressed on T cells where it plays an essential role in cell activation[1–4]. T cell activation is initiated when the T cell receptor engages with an antigen-presenting cell which in turn drives rapid calcium ion influx. Calcium entry makes the T cell membrane less negative (depolarized) which is the stimulus that opens Kv1.3 channels. Potassium ($K^+$) ions leave the cell through Kv1.3 which makes the cell membrane more negative (repolarized), and thereby sustains the large driving force for calcium entry that is needed for T cell activation and proliferation. If Kv1.3 is inhibited or unable to conduct $K^+$ ions, the T cell cannot undergo activation which in turn promotes immune suppression[5–12]. The ability to suppress the immune system via Kv1.3 motivates the development of therapeutic agents to inhibit the channel and ameliorate a number of autoimmune diseases including multiple sclerosis, type 1 diabetes, and rheumatoid arthritis[13]. The FDA-approved drug clofazimine inhibits Kv1.3 and is used for the treatment of psoriasis and graft-versus-host disease[14,15], and other small molecule, peptide, and immunoglobulin-based modulators are under development[13,16]. However, there is limited structural information about Kv1.3 interactions with its modulators which restricts our ability to rationalize their mechanisms.

Kv1.3 is one of eight members of the mammalian Kv1 family of Kv channels and the only one expressed on human T cells[1,4,17,18]. The Kv1 channels are tetramers and each subunit has a large cytoplasmic domain called T1, and six transmembrane helices called S1-S6, with S1-S4 forming a voltage-sensing domain (VSD) and S5-S6 forming the ion conducting pore domain[19–23]. The S5-S6 region harbors the P-loop, which in turn contains the highly conserved selectivity filter responsible for coordinating $K^+$ ions and maintaining $K^+$ permeation[24]. The selectivity filter contains the TXGYG motif which is highly conserved in potassium channels and is known as the potassium channel signature sequence[25]. The filter is structured with the backbone carbonyls of the signature sequence directed into the pore along the four-fold symmetry axis of the channel[26]. The structure formed by the carbonyls is evolved to precisely coordinate dehydrated $K^+$ ions and select for potassium over sodium[27]. Structures of potassium channels most commonly show the selectivity filter coordinating up to four $K^+$ ions with additional $K^+$ ions in the cavity below the filter, or situated above the filter[19,21,24,26]. The four S6 helices in a tetramer come together on the intracellular side of the membrane to form the internal gate. The gate is closed when the cell is at rest but upon membrane depolarization the VSDs undergo a conformational change that opens the gate and allows $K^+$ ion efflux[28–30].

When Kv1 channels experience sustained activation they undergo C-type inactivation, a process that diminishes ion permeation and is thought to involve a conformational change in the selectivity filter that disrupts ion coordination[23,31–41]. Recent structural work on the Kv1.2-2.1 chimera and the Shaker channel from *D. melanogaster* suggest that C-type inactivation involves disruption of two hydrogen bonds in the pore region, between Asp and Trp or between Tyr and Thr/Ser[23,41]. These structures also show dilation of the external end of the selectivity filter, resulting in partial loss of ion coordination. Indeed, molecular dynamics simulations support the notion that these conformational changes impair ion permeation[23]. The results are consistent with key functional data showing that the aforementioned hydrogen bonds in the pore region function as "molecular timers" which set the rate of C-type inactivation[40]. In addition to regulating $K^+$ influx into T cells, C-type inactivation controls cell excitability in neurons and cardiac cell repolarization by determining the number of Kv channels available to open[42]. Understanding C-type inactivation, the overall gating process of Kv1 channels, and how gating can be pharmacologically manipulated is of fundamental importance to basic physiology and drug design.

Here we aim to define the mechanisms for two recently developed immunoglobulin-based Kv1.3 modulators, a nanobody and an engineered IgG antibody. First, we solve the cryo-electron microscopy (cryo-EM) structure of human Kv1.3 without modulators as a reference. We find the selectivity filter adopts two distinct conformations which are both different from previously reported Kv1 family structures resolved in conformations thought to be conducting[19,23,26,41]. Next, we solve the structure of Kv1.3 with the recently-developed nanobody inhibitor A0194009G09 (Ablynx/Sanofi) which inhibits with nanomolar affinity through an unknown mechanism and has 1,000-fold selectivity for Kv1.3 over hERG, Kv1.5, and Kv1.6[43]. We then solve the structure of Kv1.3 with MNT-002 (Minotaur Therapeutics), an engineered antibody where the peptide toxin ShK was fused into the complementary determining region 3 (CDR3) of an IgG antibody[44]. ShK is a 35-amino acid-residue polypeptide from the sea anemone *Stichodactyla helianthus*, and is a high affinity blocker of the Kv1.3 channel in T cells that binds with a stoichiometry of one ShK per Kv1.3 tetramer[45]. ShK or its derivatives block Kv1.3 with high selectivity over other Kv channels[45,46], and show efficacy in rat models of autoimmune conditions including arthritis[5,47], diabetes[48,49], dermatitis[50], and asthma[51]. The Fc region of the antibody engages Fc receptors to enhance inhibition of T cells (manuscript in preparation). Because IgG antibodies are divalent, the antibody-ShK fusion has potentially increased avidity compared to free ShK and the fusion is expected to have enhanced half-life. Taken together these results enrich our understanding of Kv1.3 pore dynamics and channel gating, and establish mechanisms of Kv1.3 modulation by two distinct molecules.

## Results

**Structure of human Kv1.3 and its selectivity filter conformations.** Before investigating Kv1.3 in complex with modulators we first aimed to define a high-resolution reference structure of the channel in an unbound state. We expressed and purified full-length human Kv1.3 and solved the structure using single particle cryo-EM to 2.89 Å with C4 symmetry in 150 mM KCl (Fig. 1, Supplementary Fig. 1, Supplementary Table 1). Inspection of the density map showed clear densities for the VSDs, the channel pore domain and the cytoplasmic T1 domains. (Fig. 1, Supplementary Fig. 1). A local resolution heat map (Supplementary Fig. 1) shows the core of the protein is better resolved compared to the periphery. The S1-S2 and S3-S4 linkers and N-terminus (102 residues) and C-terminus (84 residues) were not resolved, likely because of local flexibility. The overall architecture of the structure is consistent with recent structures of Kv1.3[52,53], structures of Kv1.2[19,20], the Kv1.2-2.1 chimera[21,22,41], and the orthologous Shaker Kv channel from *D. melanogaster*[23]. Putative detergent densities were observed at the hydrophobic interior surface of each S6 helix but show no apparent influence on the structure (Supplementary Fig. 1). The VSDs are in the 'up' position (Fig. 1C) and the internal gate is open (Fig. 1D, Supplementary Fig. 6), reflecting the fact that in the experimental conditions the channel is at 0 mV and not at a negative voltage which would otherwise keep the channel in a resting conformation with the VSDs 'down' and the internal gate closed. Although elevated external $K^+$ reduces the speed and extent of C-type inactivation[38], for Kv1.3 most channels are inactivated at 0 mV even with elevated external $K^+$ concentrations (Supplementary Fig. 2), raising the possibility that our structure might represent a C-type inactivated conformation.

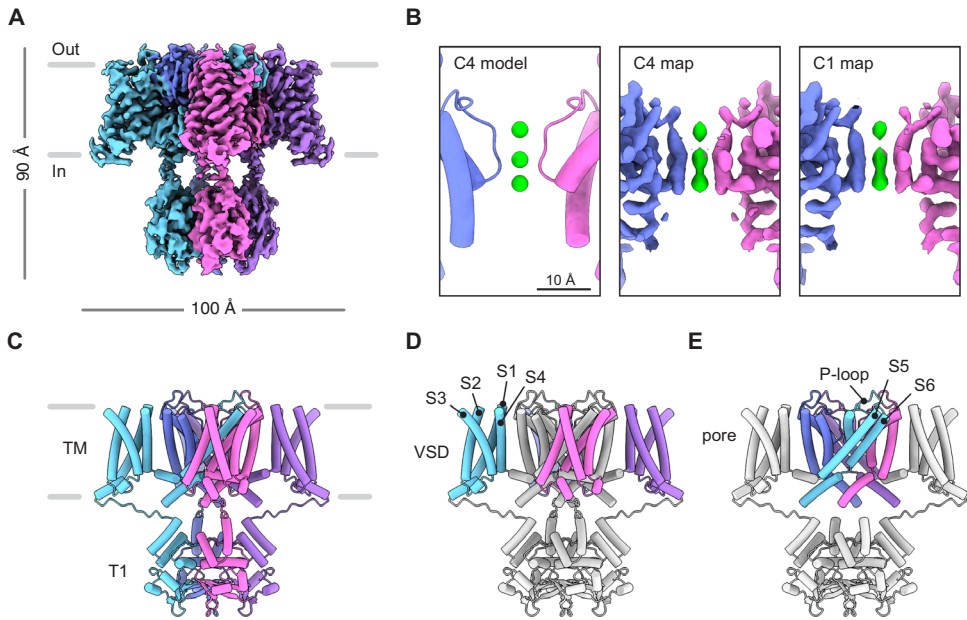

**Fig. 1 Structure of human Kv1.3. A** Cryo-EM density map of human Kv1.3. **B** The pore region of Kv1.3 as modeled with C4 symmetry (left), as seen in the density map refined with C4 symmetry (middle) or as modeled with C1 symmetry (right). This panel highlights the presence of three $K^+$ ion densities and the equivalent ion occupancy when using C1 and C4 symmetries. $K^+$ ions shown in green. **C** Model of human Kv1.3 with C4 symmetry constraints. The transmembrane (TM) region and T1 domains are indicated. Model of Kv1.3 with the voltage sensing domains (VSDs) colored (**D**), or with the pore domain colored (**E**). The front and rear VSDs are omitted for clarity in **E**.

To explore this possibility, we inspected the pore region of the cryo-EM map and evaluated the conformation of the selectivity filter (Fig. 2). We observed the P-loop appears to adopt two discernable conformations in the short region from Tyr447 to Met450 (Fig. 2A, B). The conformations were present in all four subunits even when the data was processed without imposing symmetry (C1), suggesting that our initial particle classification and structural refinement was not sensitive to this particular structural variation. These observations motivated us to obtain separate structures for each loop conformation using computational image classification, and also address whether individual tetramers are conformationally homogeneous in the loop region or if tetramers contain a combination of the two subunit conformations. We used symmetry expansion in Relion[54] to isolate each subunit within each tetramer and 3D classification to obtain density maps for the Kv1.3 subunit in both conformations (Fig. 2C, Supplementary Fig. 1). The fact that Kv1.3 is conformationally heterogeneous presented a challenge to structural modeling since a model can capture only one conformation. Given this constraint, we chose to build two tetramer models representing the two subunit conformations (i.e. each tetramer model has four subunits which are all in the same conformation).

To analyze the Kv1.3 subunit conformations we required a reference structure in an active state and selected the Kv1.2-2.1 structure[21] (Fig. 2D). This structure was chosen because of its sequence similarity to Kv1.3, and its high resolution compared to a recent Kv1.3 structure which had particularly limited resolution in the pore region[52]. Both of our Kv1.3 subunit conformations show a dilation of the selectivity filter with $K^+$ ion densities in three ion binding sites (Fig. 2E, F, H, I, Supplementary Fig. 1, Supplementary Fig. 6, Supplementary Fig. 7). In both C1 and C4 symmetry maps the three ion densities appear similar in size which suggests similar occupancies (Fig. 1B). In contrast, the active state reference structure has a narrow selectivity filter coordinating four $K^+$ ions (Fig. 2G). Accordingly we designated the two subunit conformations in our Kv1.3 structure dilated 1 (D1) and dilated 2 (D2) (Fig. 2E, F). The first dilated

conformation (D1) shows the apex of the P-loop folded outward relative to the reference structure (Fig. 2G, H, Supplementary Fig. 6, Supplementary Fig. 7). In this conformation, Asp449 of Kv1.3 is directed upward and away from the channel, and in particular is displaced from Trp436 (Fig. 2K). This is significant because in the active state structure the corresponding Asp (Asp375) orients downward with its carboxylate moiety positioned near the indole nitrogen of Trp (Trp362) (Fig. 2J). The second dilated conformation (D2) shows a more severe disruption of the selectivity filter (Fig. 2I). Asp449 is displaced from Trp436 in D2, just as in D1, but Asp449 adopts an entirely different position in D2 (Fig. 2L). In addition, Tyr447 points away from Thr441 (Fig. 2O) which is important because these two residues are hydrogen bonding partners in the active reference structure (Fig. 2M) and the D1 conformation (Fig. 2N). The Asp-Trp and Tyr-Thr hydrogen bonds are highly conserved structural motifs in Kv channels, and rupture of the bonds is thought to accompany C-type inactivation in the Shaker Kv channel[35,38,40,55,56]. Although increasing the external $K^+$ concentration does diminish the extent of inactivation, a sizable fraction of channels remain inactivated at 100 mM external $K^+$ and at 0 mV (Supplementary Fig. 2), conditions similar to those used for structure determination. Thus, the D2 conformation we observed is likely to represent an inactivated state principally because it exhibits reduced ion coordination compared to Kv channel structures which are thought to represent conductive states, and it shows rupture of two hydrogen bonds which are known to control entry into the C-type inactivated state[40]. Also significant is that the D2 conformation of the Kv1.3 selectivity filter resembles structures of mutants of the *D. melanogaster* Shaker Kv channel[23] and the Kv1.2-2.1 chimera[41] which are proposed to be C-type inactivated conformations (Supplementary Fig. 7). The D1 state may be an intermediate between the inactivated state and the conducting state. Importantly, we note our ability to restore a conformationally homogeneous, active-like conformation of the channel by adding Fab-ShK (see below).

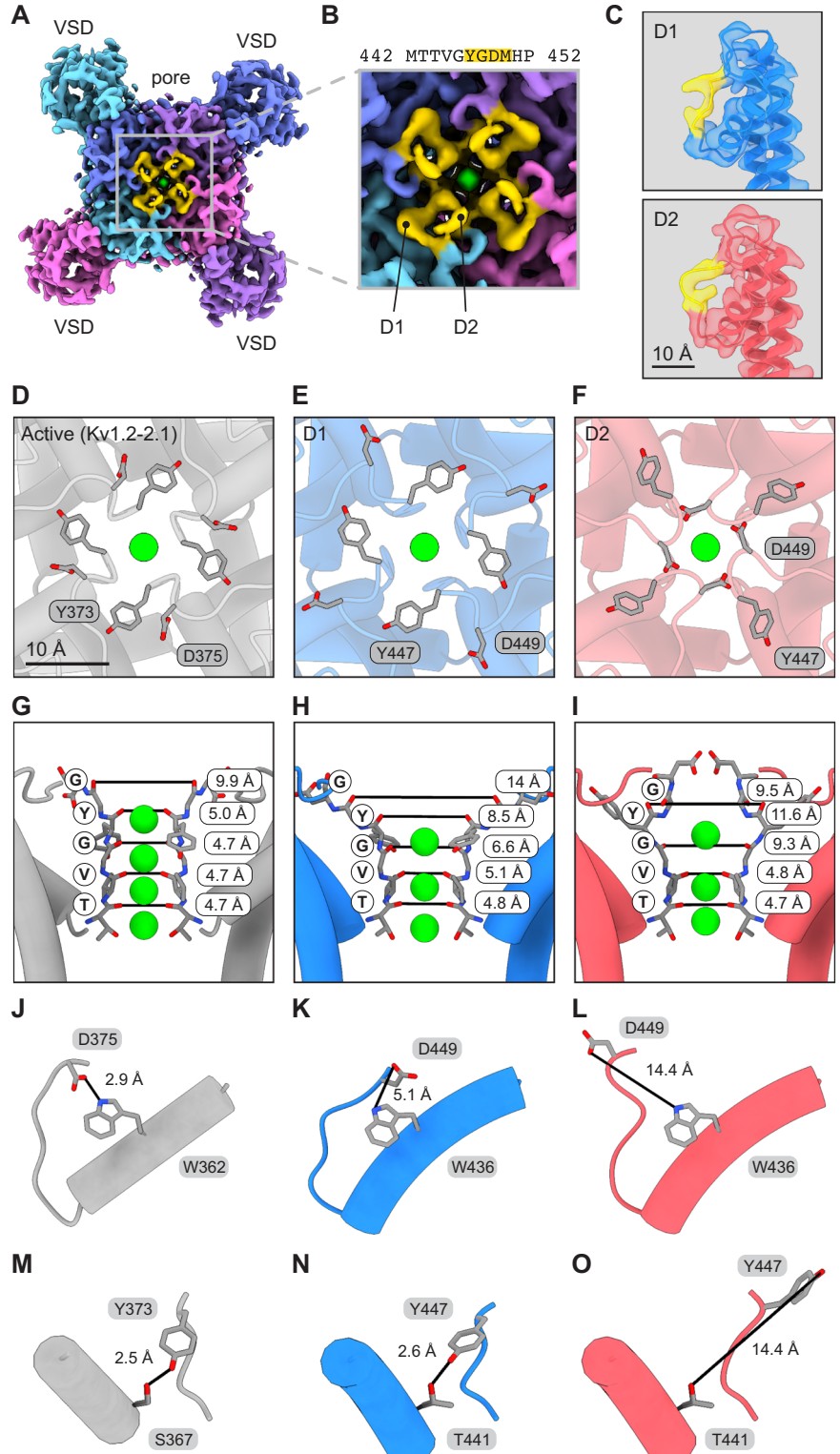

**Fig. 2 The Kv1.3 selectivity filter adopts two distinct dilated conformations. A, B** Kv1.3 cryo-EM map as viewed from the extracellular space with the D1 and D2 loop regions highlighted in yellow. **A** shows the entire extracellular surface of the channel and **B** shows a magnified view of the pore region. **C** Cryo-EM maps for individual D1 and D2 subunits as determined by symmetry expansion and 3D classification. The maps are shown with models to highlight the different selectivity filter conformations. The region from Tyr447 to Met450 is highlighted in yellow. **D–I** Views of the pore for active state (Kv1.2-2.1, PDB 2R9R (Kv1.2-Kv2.1 structure)), D1 conformation, and D2 conformation. The pores are shown from the extracellular space (**D–F**) or from the side perspective (**G–I**). Distances between carbonyl oxygens of the TVGYG motif are reported and marked with black lines. **J–O** Asp and Trp residues (**J–L**) and Tyr and Ser/Thr residues (**M–O**) involved in hydrogen bonds that regulate C-type inactivation. Distances between hydrogen bonding partners are reported and marked with black lines. K$^+$ ions are shown in green in panels **D–I**.

The 3D subunit classification analysis provided conformational identities of individual subunits in the dataset, which in turn enabled a mapping of these identities back to the parent tetramers (Supplementary Fig. 1, Supplementary Table 2). In this way we determined the conformational composition of each tetramer. We excluded from the analysis those tetramers for which one, two, or three subunits could not be classified. This left 6,598 tetramers where all four subunits could be identified as either D1 or D2 (Supplementary Table 2). Out of these tetramers, the majority were a mixture of D1 and D2 subunits with 1:3 (D1:D2) being the most abundant stoichiometry (30.8% of tetramers), followed by 2:2 (24.5%) and 3:1 (13.6%) (Supplementary Table 2). The remaining tetramers were either pure D1 having a 4:0 ratio (4.5%) or pure D2 with a 0:4 ratio (26.6%) (Supplementary Table 2). Together this analysis showed that the vast majority of tetramers analyzed are a mixture of D1 and D2 subunits, suggesting that the D1 and D2 filter conformational changes do not occur in a fully cooperative manner. However, it is essential to consider that this analysis included 43% of the Kv1.3 subunits (324,390/763,172), while the remaining subunits were excluded because they could not be classified (Supplementary Table 2). In addition, the final set of tetramers analyzed for their composition account for only 3.5 % of the total tetramer population (6,598/190,793) (Supplementary Table 2). As such, the precise D1:D2 ratios we determined for the subset may not be representative of the dataset as a whole.

**The nanobody binds and bridges the voltage sensing and pore domains**. We first validated the function of purified A0194009G09 nanobody (NB) (Fig. 3A) by applying it to cells expressing Kv1.3 and measuring $K^+$ currents in physiological external solutions containing low $K^+$ (Fig. 3B, C). These current recordings showed that under control conditions, Kv1.3 was activated by membrane depolarization and displayed its characteristic slow C-type inactivation. In contrast, when NB (1 nM to 1 μM) was added to the extracellular solution Kv1.3 channel inactivation was greatly accelerated and current was almost completely inhibited by the end of the test depolarizations (Figs. 3B, C, 4A, B). Importantly, acceleration of inactivation by the NB was also observed in solutions containing high external $K^+$ and the channel inactivates nearly completely following relatively short depolarizations to 0 mV in the presence of the NB (Supplementary Fig. 2). To solve the structure of Kv1.3 with the NB inhibitor the proteins were combined in a 3:1 molar ratio (3 NB per Kv1.3 subunit) and imaged by single particle cryo-EM. The data was initially processed with C1 symmetry and four copies of the NB were observed on the extracellular surface of Kv1.3 (Supplementary Fig. 3). This result supported the use of C4 symmetry to process the data and a density map was refined to 3.25 Å resolution followed by model building (Fig. 3D–F, Supplementary Fig. 3). A heat map visualizing the local resolution throughout the density map shows a well resolved protein core with slightly attenuated resolution at the periphery (Supplementary Fig. 3). Resolution is highest in the channel region and lowest at the top of the nanobodies. Three $K^+$ ion densities are visible in the pore and have similar occupancies (Fig. 3G). The S1-S2 linkers were not resolved and, although the S3-S4 linkers were not fully resolved, they had improved resolution compared to the unbound structure, with an additional nine residues which could be modeled. This region is in close proximity to, and is likely stabilized by, the NB (Fig. 3L).

In the pore region of the NB-bound structure Asp449 is flipped away from Trp436 in an orientation incompatible with hydrogen bond formation (Fig. 3H). However, Tyr447 adopts two different rotamers in the density map. One Tyr447 rotamer is similar to

the D1 conformation of unbound Kv1.3, with its hydroxyl group directed towards Thr441 (Fig. 3I, Supplementary Fig. 7). The second Tyr447 rotamer has its hydroxyl group directed towards the extracellular space and so it is distinct from both D1 and D2 (Fig. 3I, Supplementary Fig. 7). For this reason, we designated the conformation with the second Tyr447 rotamer as D3. Thus, when Kv1.3 is bound by the nanobody the selectivity filter adopts D1 and D3 conformations.

The nanobodies do not make direct contact with the selectivity filter of Kv1.3, but rather they decorate the turret loops near the pore and extend contacts to the VSDs (Fig. 3F). On the turret loop, Phe103 of the NB forms a hydrophobic interaction with Ala421, Arg100 forms a hydrogen bond with Ser426, and Tyr104 and Pro424 make hydrophobic contact (Fig. 3K). Arg32 of the NB extends directly over Gly427 of the turret loop. This glycine is notable because all other human Kv1 channels display large side chains (His, Gln, or Leu) at the equivalent positions, which may clash with Arg32 (Fig. 3K, M). Arg51 and Asn59 of the NB both appear capable of hydrogen bonding with Tyr265 on the VSD, while Trp99 of the NB forms a hydrophobic interaction with Pro266 (Fig. 3L, N).

The structure shows four NBs can bind the Kv1.3 tetramer (Fig. 3D, E, Supplementary Fig. 3A) but it was solved with a three-fold molar excess of NB (three NB per Kv1.3 subunit). This motivated us to investigate the concentration-dependence for inhibition of Kv1.3 by the NB. First, we measured the rate of inactivation in control solutions and in the presence of the NB at concentrations ranging from 1 nM to 1 μM. Inactivation in the absence of NB could be described by a double exponential function and at the lowest concentration of 1 nM, the NB did not appreciably alter these time constants but increased the amplitude of the faster component (Fig. 4A, B). For example, at +50 mV in control solution the amplitude of the fast component was 19%, whereas at 1 nM NB its amplitude increased to 51%. With 10 nM NB inactivation was accelerated and was modeled again with multiple exponential functions, with the faster component having an amplitude of 94.5% at +50 mV (Fig. 4A, B). At the highest NB concentrations of 100 nM and 1 μM, inactivation was dramatically accelerated compared to control solutions (Fig. 4A, B). Although we could not resolve unique kinetic components of inactivation corresponding to distinct populations of Kv1.3 channels with all possible stoichiometries between NB and channel, the components we could resolve and their dependence on NB concentration reveal the presence of multiple distinct occupancies. Second, we examined the concentration dependence for the onset and the recovery from inhibition by the NB at 10 and 100 nM. Onset of inhibition was sped by increasing the concentration of the NB from 10 nM to 100 nM (Fig. 4C, E). In contrast, recovery from inhibition was slowed by increasing the NB concentration. This was inferred from the sigmoidal recovery profile following removal of the NB, and the increase in this sigmoidicity with higher NB concentration (Fig. 4D, E). Taken together, these functional results show that NB binding occupancy on Kv1.3 can vary and that higher occupancies promote faster and more durable inactivation.

**ShK toxin block induces a conductive pore conformation**. To investigate the MNT-002 antibody-ShK fusion, we first generated the Fab-ShK form of the molecule (Fig. 5A). This was done because the antibody-ShK is divalent and would likely crosslink two Kv1.3 molecules together. In contrast, the monovalent Fab-ShK can bind only one Kv1.3 channel, thereby forming a protein complex amenable to single particle cryo-EM structure determination. We validated the ability of Fab-ShK to block Kv1.3 currents with high affinity at concentrations ranging from 0.1 nM to

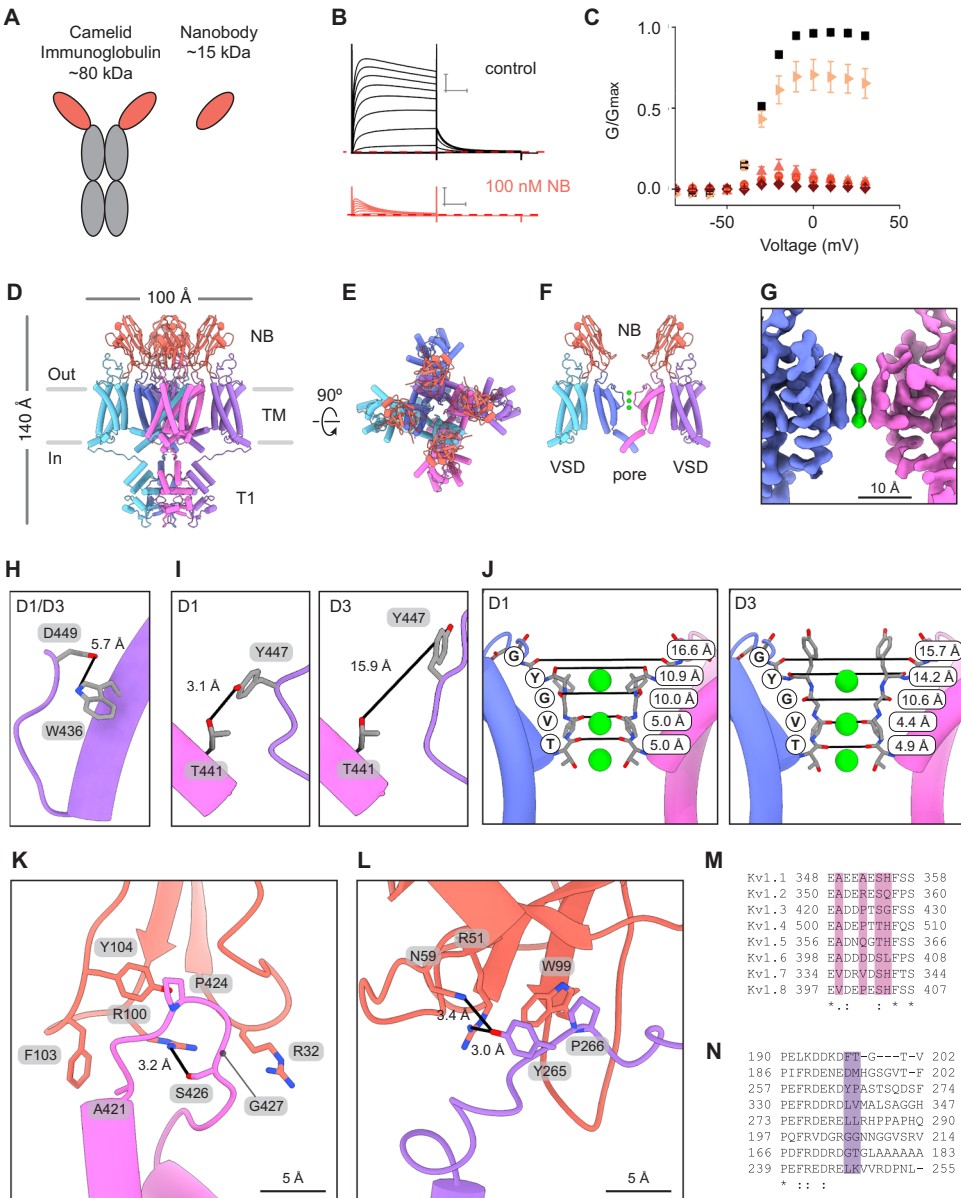

**Fig. 3 Multiple nanobodies bind the turret loops and voltage sensing domains of Kv1.3. A** Illustration of camelid antibody and nanobody. **B** Kv1.3 current traces obtained from a family of depolarizing pulses ranging from −80 mV to +30 mV in control 2 mM K$^+$ solutions (black) and in the presence of 100 nM nanobody (NB) A0194009G09 (orange). Holding voltage was −80 mV and tail voltage was −50 mV. Red dotted line denotes zero current level. Scale bars indicates 2 µA and 50 ms. **C** G-V relations obtained in control 2 mM K$^+$ solutions (black squares; $n = 20$), 1 nM (right facing triangles, light orange; $n = 7$), 10 nM (upright triangles, pink; $n = 3$), 100 nM (circles, orange; $n = 7$) or 1 µM nanobody (diamonds, dark red; $n = 3$) by measuring the peak of the tail current at −50 mV and normalizing it to the maximum tail current in control solution. Error bars are SEM. Side (**D**) and extracellular (**E**) views of the model for Kv1.3 in complex with four nanobodies (orange). **F** Cutaway view of the structure in (**D**) showing two nanobodies with each bound to a pore domain and a VSD of Kv1.3. The T1 domains are omitted for clarity. **G** Cryo-EM density of the pore region (C1 symmetry). **H**, **I** Views of W436/D449 and T441/Y447 for both the D1 and D3 conformations identified in the Kv1.3-nanobody structure. **J** Structures of the selectivity filter of the Kv1.3-nanobody complex with the D1 (left) and D3 (right) conformations. Distances between carbonyl oxygens of the TVGYG motif are reported and marked with black lines. K$^+$ ions shown in green. Close up views of a nanobody in complex with a turret loop of a pore domain (**K**) and the S1-S2 loop of a VSD (**L**). Sequence alignment for the turret loop (**M**) and VSD S1-S2 loop (**N**) regions for human Kv1 channels. Source data are provided as a Source Data file.

10 nM in external solutions containing low external K$^+$ (Fig. 5B, C). At a concentration of 10 nM, Fab-Shk completely inhibited Kv1.3 currents recorded with either low or high external K$^+$ concentrations (Supplementary Fig. 4E, 4F) and the extent, onset and recovery from inhibition were similar to those observed with ShK alone (Supplementary Fig. 4A–D). We then solved the structure of Kv1.3 with Fab-ShK to a global resolution of 3.39 Å (Supplementary Fig. 5). The local resolution heat map of the structure shows the best resolution is in the channel region and

that the T1 domains have somewhat attenuated resolution (Supplementary Fig. 5). The S1-S2 and S3-S4 linkers were not resolved likely owing to intrinsic flexibility. The ShK portion of the Fab-ShK molecule was resolved to ~3 Å resolution. In contrast, the local resolution of the Fab region was estimated to be ~15 Å (a precise resolution could not be calculated with confidence). We attributed the low resolution of the Fab region to flexibility in the CDR3 'stalk' connecting ShK and the Fab. The density map enabled atomic modeling of Kv1.3 and the ShK toxin

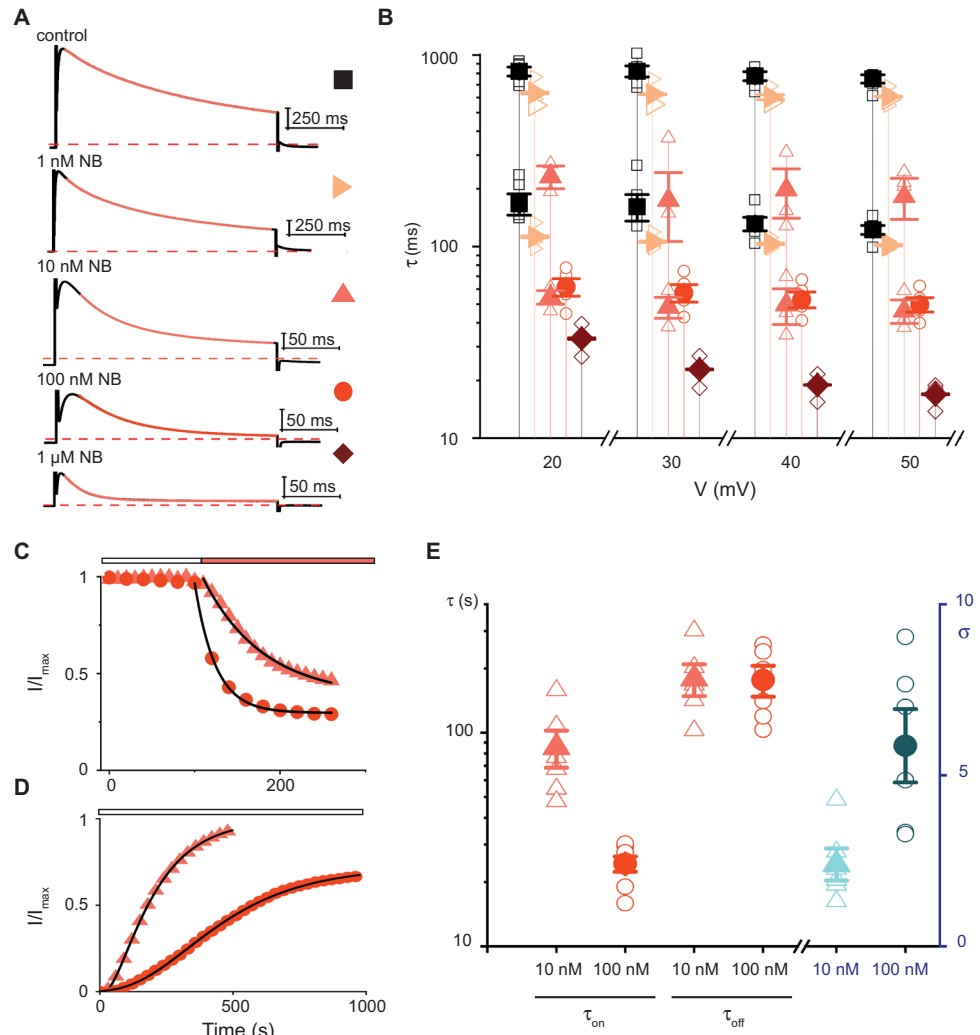

**Fig. 4 Concentration-dependence for acceleration of inactivation by the nanobody. A** Kv1.3 current traces elicited by depolarization to +50 mV in 2 mM K$^+$ in the absence or presence of the NB. Holding voltage was −80 mV, tail voltage was −50 mV, current scale bars are 1 µA and red dotted line is zero current. Control trace and 100 nM NB are from one cell, and traces for other NB concentrations are from three different cells. Orange lines are fits of mono- (100 nM and 1 µM) or double exponential (control, 1 nM and 10 nM) functions (see Methods). **B** Influence of the NB on inactivation. Mean values for $\tau_{slow}$ and $\tau_{fast}$ are shown for control (filled black squares; $n = 5$;), 1 nM NB (right facing light orange triangles; $n = 3$) or 10 nM NB (filled upright pink triangles; $n = 3$). Mean values of $\tau$ are shown for 100 nM NB (filled orange circles; $n = 5$) and for 1 µM NB (filled dark red diamonds; $n = 3$). Open symbols denote individual experiments. Error bars are S.E.M. **C** Onset of inhibition by NB in 2 mM K$^+$ solutions. Peak currents elicited by 500 ms steps from −80 to +40 mV were normalized to the maximum value in control. Addition of NB (orange bar) at 10 nM (triangles) or 100 nM (circles). Black lines are fits of a single exponential function to the data (see Methods). **D** Recovery from inhibition by the NB in 2 mM K$^+$ solutions. Same voltage protocol as **C** with removal of the NB indicated by white bar (10 nM NB triangles and 100 nM circles). Currents were scaled by setting the initial values in NB to zero and normalizing to the maximum before the addition of the NB. Pulses delivered every 10 s but shown every 30 s for clarity. Black lines are fits of a sigmoidal function to the data (see Methods). **E** Concentration dependence for onset ($\tau_{on}$) and recovery ($\tau_{off}$) for NB. Mean values (filled symbols) for $\tau_{on}$ ($n = 6$ for 10 nM and 7 for 100 nM), $\tau_{off}$ and σ ($n = 6$), individual values (open symbols) from experiments like those in **C**, **D**. Error bars are S.E.M. Source data are provided as a Source Data file.

(Fig. 5D, Supplementary Fig. 5) but the Fab region was modeled using a more conservative approach by generating a Fab homology model (PDB 6OO0 (bovine Fab NC-Cow1 structure)) and rigid body fitting the homology model into the density map (Fig. 5D, Supplementary Fig. 5).

The Fab binds with an ~45° angle to the extracellular surface of Kv1.3 and its long and narrow CDR3 'stalk' positions ShK at the mouth of the pore (Fig. 5D). The Fab portion of Fab-ShK makes no contact with the extracellular loops of Kv1.3, or any other part of the channel. ShK forms its most significant contact with the pore via Lys22, with the ammonium group hooked downward into the selectivity filter to coordinate the backbone carbonyl of Tyr447 on all four Kv1.3 subunits (position TVGYG of the K$^+$

channel signature sequence) (Fig. 5E–G, Supplementary Fig. 6F). In this way Lys22 functions as a surrogate K$^+$ ion as proposed[57–60]. We compared the conformation of the selectivity filter in the Kv1.3 with Fab-ShK structure to the active state reference structure and the D1 and D2 conformations observed in unbound Kv1.3 (Fig. 2, Supplementary Fig. 7). This showed that when the selectivity filter is bound by ShK it adopts a similar conformation as seen in active K$^+$ channel structures[21,23,41] (Figs. 2, 5E–H). Despite the absence of the outermost K$^+$ ion in the filter, the key hydrogen bonds formed by Asp449/Trp436 and Y447/T441 are intact (Fig. 5I, J), consistent with an active channel conformation (Fig. 2J, M). Thus, ShK stabilizes an active conformation of the selectivity filter despite blocking ion flow.

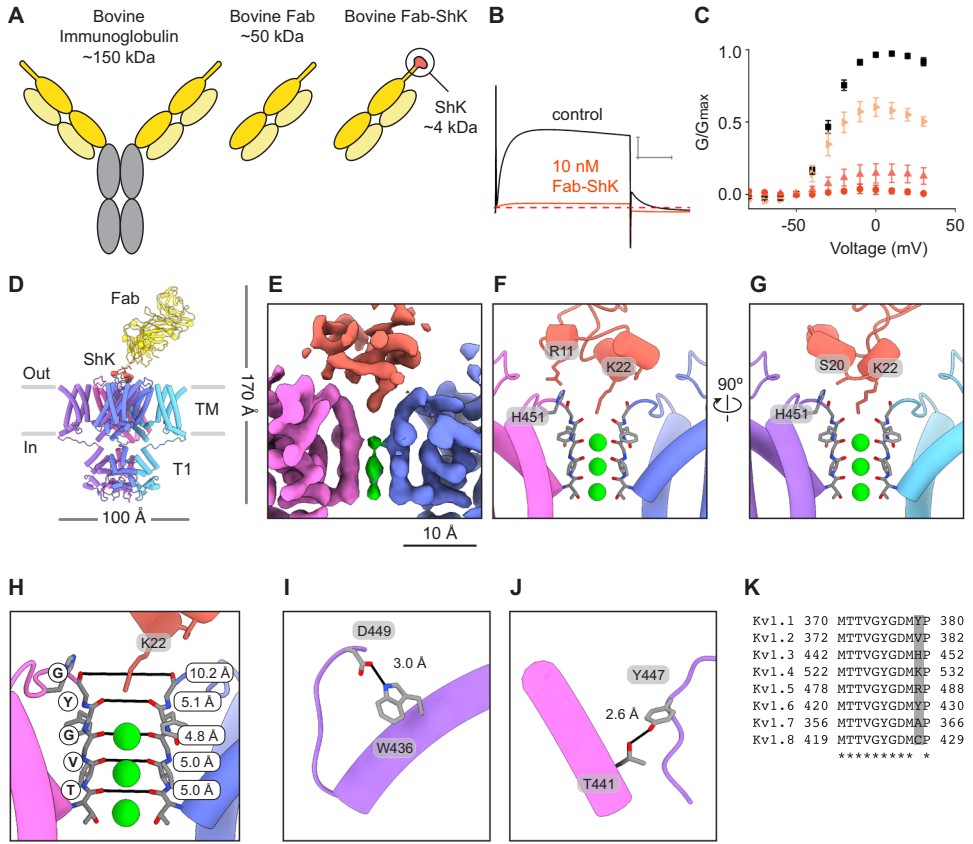

**Fig. 5 Fab-ShK stabilizes a conductive pore conformation to block the channel. A** Illustration of bovine antibody which has an elongated CDR3 region compared to human antibodies, bovine Fab fragment, and Fab-ShK fusion. **B** Kv1.3 current traces obtained by depolarizations to 0 mV from control 2 mM K+ solutions (black) and in the presence of 10 nM Fab-ShK (orange). Holding voltage was −80 mV and tail voltage was −50 mV. Red dotted lined denotes zero current level. Scale bar indicates 5 μA and 50 ms. **C** G-V relations obtained in control 2 mM K+ solutions (black squares; $n = 15$), 0.1 nM (right facing light orange triangles; $n = 4$), 1 nM (upright pink triangles; $n = 5$) or 10 nM Fab-ShK (orange circles; $n = 6$) by measuring the peak of the tail current at −50 mV and normalizing it to the maximum tail current in control solution. Error bars are SEM. **D** Model of Kv1.3 in complex with Fab-ShK as viewed from the side, with Fab and ShK colored yellow and orange, respectively. Close up views of ShK docking at the selectivity filter of Kv1.3. The map is displayed in **E** and the two views of the model in **F**, **G**. The ShK residue numbers correspond to ShK alone, not their numbering in the Fab-ShK fusion. The residues Arg11, Ser20, and Lys22 of ShK correspond to Arg118, Ser127, and Lys129 in the Fab-ShK fusion. **H** Structure of the selectivity filter of the Kv1.3-ShK complex. Distances between carbonyl oxygens of the TVGYG motif are reported and marked with black lines. **I**, **J** Key hydrogen bonds involved in channel gating are stabilized when Kv1.3 is bound by ShK. **K** Sequence alignment for the selectivity filters of Kv1 channels with the position for His451 of Kv1.3 highlighted. K+ ions are shown in green in panels **E**–**H**. Source data are provided as a Source Data file.

In addition to the contact between Lys22 and the selectivity filter, Ser20 and Arg11 of ShK form hydrogen bonds with His451 of Kv1.3, but on separate subunits (Fig. 5F, G). This is significant because Ser20 and Arg11 are major determinants of ShK toxin binding[60] and because Kv1.3 is the only family member with a histidine in this position, with most other Kv1 channels having residues that would be less suited to forming the contact with Ser20 because of charge or steric mismatch (Fig. 5K).

## Discussion

In this study we determined the mechanisms of Kv1.3 modulation for a recently-developed nanobody inhibitor and an engineered fusion of an IgG antibody with the ShK toxin. We first solved a structure of unbound Kv1.3 which showed a dynamic selectivity filter with two discernable conformations (D1, D2) (Figs. 1, 2). The dilation of the filter in D1 and D2 reorients the critical backbone carbonyls to disrupt K+ ion coordination, a feature which has been proposed to be associated with C-type inactivation[23,41] and may serve to increase the energetic barrier to K+ permeation[34]. The dilated conformations also show disruption of two key hydrogen bonds which are thought to stabilize an active channel conformation (Fig. 2)[23,35,38,40,41,55,56,61].

Because D1 and D2 exhibit progressive rupture of these two hydrogen bonds (Fig. 2), we hypothesize that they may correspond to partially inactivated and inactivated states, respectively.

Structures of Kv1.3 were recently reported by Tyagi et al. in an unbound state and in complex with the blocker dalazatide which is a derivative of ShK[53]. The unbound structure (PDB 7WF3 (structure of Kv1.3 channel in apo state)) shows a dilated selectivity filter that closely matches the D2 conformation reported in the present study (Supplementary Fig. 7). However, the study does not report observing a second conformation such as D1, as we observe in our investigation. It is unknown why this is so, but we offer some points of consideration. Both Kv1.3 structures were solved using cryo-EM with protein solubilized in a mixture of DDM and CHS, and in the presence of 150 mM KCl, so the biochemical conditions do not account for structural differences. One salient difference is that in the Tyagi study the protein was truncated by 52 residues at its N-terminus whereas in the present study we used full-length protein. A second difference is that Tyagi and coworkers solved the structure in complex with a beta subunit which is known to modify Kv1.3 function by increasing current amplitude[62]. Third, the Tyagi study used Sf9 expression whereas we used HEK expression. It is also possible that a second

pore loop conformation was present in the Kv1.3 structure from the Tyagi study, but that it was not detected during image classification. Overall, the attributes of the Kv1.3 structure we report are consistent with the biochemical equilibrium conditions used in our cryo-EM experiments (0 mV for many hours), and a model for C-type inactivation which was proposed from functional experiments[35–38] and which is coming into sharper focus from recent structural work on other Kv channels[23,41].

The A0194009G09 nanobody-bound structure reveals an unanticipated mechanism of channel inhibition in which the nanobody bridges the turret loops and the VSDs to promote C-type inactivation (Figs. 3, 4). The structure has two distinct conformations (D1, D3) distinguished by different rotamers for Tyr447 which is a critical residue in C-type inactivation[40]. It is notable that the regions where the nanobody binds are among the most variable segments in the Kv1 channel family (Fig. 3). Atomic delineation of the nanobody interaction with Kv1.3, along with increasing availability of potassium channel structures, opens the possibility of designing new nanobodies with high specificity for different channels. The FDA approval in 2019 of the first nanobody drug[63] and the exponential rise of FDA-approved immunoglobulins[64] suggests this may be a productive direction for future work.

Last, we solved the structure of Kv1.3 with the Fab form of the MNT-002 antibody. Our structure is consistent with the recently reported dalazatide-bound Kv1.3 structure (Supplementary Fig. 7) (PDB 7WF4 (structure of Kv1.3 channel in dalazatide-bound state))[53] and shows the ShK toxin switches the selectivity filter from the D1 and D2 conformations present in the unbound structure to an active conformation, yet blocks the ion permeation pathway (Fig. 5). ShK is a prominent member of a large family of sea anemone toxins that bind many K$^+$ channels[65] so the structure serves as a template to understand these toxins. In addition, many animal venom toxins block K$^+$ channels and have been extensively studied, but the structural details of these interactions have been challenging to fully define owing to methodological limitations[57]. The venom toxins, like ShK and the sea anemone blockers, employ a highly conserved lysine residue to block the channel. Thus, the details of our structure may inform the understanding of channel block for many classes of toxins and potassium channels.

## Methods

**Expression and purification of human Kv1.3.** The gene for human Kv1.3 was cloned into the pEZT-BM BacMam expression vector[66] and at the C-terminus was fused to a 3C protease recognition site followed by the mVenus fluorescent protein and a Twin-Strep affinity tag. The expression construct was transformed into DH10Bac cells to produce bacmid which was then transfected into Sf9 cells grown in ESF 921 media (Expression Systems). P1 and P2 virus production was monitored using GFP fluorescence from the pEZT-BM vector until virus harvesting. HEK293S GnTI⁻ cells (ATCC CRL-3022) were grown (3.2 L or 6.4 L) at 37 °C and 8% CO₂ to a density of $3.5 \times 10^6$ cells/mL in FreeStyle suspension media (Gibco) supplemented with 2% fetal bovine serum (Gibco) and Anti-Anti (Gibco). P2 virus was added to cells, the suspension was incubated at 37 °C for 24 h, then sodium butyrate (Sigma) was added to a final concentration of 10 mM and flasks were shifted to 30 °C and 8% CO2. Cells were collected 84 h after transduction by low-speed centrifugation, flash-frozen in liquid nitrogen, and stored at −80 °C.

Cell pellets were resuspended in ice-cold resuspension buffer containing 50 mM Tris (pH 7.5), 150 mM KCl, 2 mM DTT, protease inhibitor tablet (Sigma), 1 mM PMSF, 0.5 mM EDTA, and 25 μg/ml DNAse (6 mL buffer per 1 g of cell pellet) and manually pipetted until no clumps remained. The cells were disrupted on ice using a QSonica Q700 sonicator (3 × 15 sec, power level 60), lysate was clarified by low-speed centrifugation (7,200 × g for 20 min), and membranes were isolated by ultracentrifugation at 125,000 × g for 2 h. Membranes were resuspended and homogenized in buffer 20 mM Tris (pH 7.5), 150 mM KCl, protease inhibitor tablet (Sigma), 1 mM PMSF, and 0.5 mM EDTA. The protein was extracted by adding 50 mM n-Dodecyl-β-D-maltopyranoside with 0.25% cholesteryl hemisuccinate (Anatrace, D310-CH210) and nutating for 1 hr at 4 °C. The mixture was ultracentrifuged at 125,000 × g for 50 min to remove insoluble material. The supernatant was filtered through a 0.45 μm filter then bound to a 5 mL Strep-Tactin column (GE) equilibrated with running buffer containing

20 mM Tris (pH 7.5), 150 mM KCl, 0.5 mM EDTA, and 2 mM DDM with 0.01% CHS. The bound protein was washed with running buffer containing 10 mM MgCl₂, 0.01% glucose, and kanamycin (25 μg/ml). When the cells reached OD₆₀₀ of 0.7, 1 mM IPTG was added to induce protein expression. After 3 h the cells were harvested and pellets stored at −80 °C. To purify the protein, cell pellets were thawed and added to buffer containing 20 mM Tris (pH 7.5), 150 mM KCl, 1 mM PMSF, 5 mM MgCl₂, 0.05 mg/mL DNAse and 0.2 mg/mL lysozyme. The mixture was stirred at 4 °C for 30 min then sonicated on ice. Lysate was clarified by centrifugation (F14-14x50cy rotor, 24,676 × g, 20 min, 4 °C) and then filtering through a 0.45 μm syringe filter. Affinity purification was done using a running buffer with 20 mM Tris (pH 7.5) and 150 mM KCl. TALON cobalt resin (Takara) was washed with water then equilibrated with 10 column volumes of running buffer. Clarified lysate was loaded on the column, washed with 10 column volumes of running buffer supplemented with 5 mM imidazole, then protein was eluted with running buffer supplemented with 50 mM imidazole. The nanobody was further purified using gel filtration with buffer containing 20 mM Tris (pH 7.5), 150 mM KCl, 0.5 mM EDTA. Peak fractions were then pooled and concentrated.

**A0194009G09 nanobody expression and purification.** The gene sequence for nanobody A0194009G09 was obtained from patent CA2951443A1[43], codon optimized for *E. coli* expression, then synthesized and cloned into the pET26b(+) vector in frame with an C-terminal 6× His tag (GenScript). BL21 DE3 cells were transformed with the plasmid and grown at 37 °C in TB media supplemented with 1 mM MgCl₂, 0.01% glucose, and kanamycin (25 μg/ml). When the cells reached OD₆₀₀ of 0.7, 1 mM IPTG was added to induce protein expression. After 3 h the cells were harvested and pellets stored at −80 °C. To purify the protein, cell pellets were thawed and added to buffer containing 20 mM Tris (pH 7.5), 150 mM KCl, 1 mM PMSF, 5 mM MgCl₂, 0.05 mg/mL DNAse and 0.2 mg/mL lysozyme. The mixture was stirred at 4 °C for 30 min then sonicated on ice. Lysate was clarified by centrifugation (F14-14x50cy rotor, 24,676 × g, 20 min, 4 °C) and then filtering through a 0.45 μm syringe filter. Affinity purification was done using a running buffer with 20 mM Tris (pH 7.5) and 150 mM KCl. TALON cobalt resin (Takara) was washed with water then equilibrated with 10 column volumes of running buffer. Clarified lysate was loaded on the column, washed with 10 column volumes of running buffer supplemented with 5 mM imidazole, then protein was eluted with running buffer supplemented with 50 mM imidazole. The nanobody was further purified using gel filtration with buffer containing 20 mM Tris (pH 7.5), 150 mM KCl, 0.5 mM EDTA. Peak fractions were then pooled and concentrated.

**MNT-002 Fab expression and purification.** The MNT-002 antibody was generated by inserting the gene for the ShK peptide into the β-ribbon 'stalk' of the ultralong CDR3 scaffold of a humanized bovine IgG as described in patent US10640574[44]. The Fab version of the antibody fusion (Fab-ShK) was expressed in freestyle HEK293 (293 F) cells (Thermo Fisher, R79007) following the manufacturer's protocol. In brief, 293 F cells were seeded at a density of $1.0 \times 10^6$ per ml prior to transfection. Cells were transfected with 293fectin (Thermo Scientific) combined with pFUSE-based plasmids (InvivoGen) encoding both heavy (HC) and light chains (LC) (HC:LC ratio of 1:1). Cell culture supernatant containing expressed protein was collected five days post-transfection, filtered through a 0.45 μm PES filter and concentrated at 4 °C using an Amicon Ultra-15 centrifugal filter unit (molecular weight cut-off of 10,000 Da) (Millipore Sigma). Protein was further purified using CaptureSelect CH1-XL affinity resin (Thermo Scientific) following the manufacturer's protocol. Purified protein was eluted in 50 mM sodium acetate (pH 4.0), then concentrated and buffer-exchanged to 20 mM Tris (pH 7.5) with 150 mM KCl and 0.5 mM EDTA at 4 °C using Amicon Ultra-4 centrifugal filter unit (molecular weight cut-off of 10,000 Da) (Millipore Sigma). The Fab yield was quantified by measuring absorbance at 280 nm on a Nanodrop One (Thermo Scientific) and evaluated by SDS-PAGE under reducing and non-reducing conditions.

**Electrophysiology.** The cDNA encoding the Kv1.3 expression construct used for protein production and cryo-EM was cloned into a pGEM vector[68], linearized with NheI and transcribed by using a mMESSAGE mMACHINE™ T7 Transcription Kit (Invitrogen/ThermoFisher). Female *Xenopus laevis* animals were housed, and surgery was performed according to the guidelines of the National Institutes of Health, Office of Animal Care and Use (OACU) (Protocol Number 1253–18). Oocytes were removed surgically and incubated with agitation for 1 hr in a solution containing (in mM) 82.5 NaCl, 2.5 KCl, 1 MgCl₂, 5 HEPES, pH 7.6 (with NaOH), and collagenase (1.2 mg/ml; Worthington Biochemical, Lakewood, NJ). Defolliculated oocytes were injected with 50 nl of channel RNA (~500 ng/ml) and maintained at 17 °C for 1–3 days in an ND96 oocyte maintenance buffer, containing (in mM): 96 NaCl, 2 KCl, 5 HEPES, 1 MgCl₂ and 1.8 CaCl₂ plus 50 mg/ml gentamycin, pH 7.6 with NaOH. Voltage-clamp recordings were performed using the two-electrode voltage-clamp recording techniques (OC-725C; Warner Instruments) with a 150 μl recording chamber that was perfused continuously. Data were filtered at 1–3 kHz and digitized at 10 kHz using a Digidata 1321 A interface board and pCLAMP 10 software (Axon; Molecular Devices). Microelectrode resistances were 0.2–0.8 MΩ when filled with 3 M KCl. For recording macroscopic Kv channel currents with low external K$^+$, the external recording solution contained (in mM): 2 KCl, 98 NaCl, 5 HEPES, 1 MgCl₂, and 0.3 CaCl₂ pH 7.6, with NaOH. For recording macroscopic Kv channel currents with high external K$^+$, the external recording solution contained (in mM): 100 KCl, 5 HEPES, 1 MgCl₂, and 0.3 CaCl₂, pH 7.6, with NaOH. Fab-ShK or the nanobody were added to the control recording solution to the indicated concentrations. All experiments were performed at room temperature (22 °C). Leak and background conductances were subtracted for tail current measurements by arithmetically deducting the end of the tail pulse of each

analyzed trace. In most instances, Kv channel currents shown are non-subtracted. Where indicated, a P/−4 leak subtraction protocol was employed[69].

G-V relationships were obtained by measuring tail currents at −50 mV and a single Boltzmann function was fit to the data according to:

$$I/I_{max} = \left(1 + e^{-zF(V-V_{1/2})/RT}\right)^{-1} \quad (1)$$

where z is the equivalent charge, $V_{1/2}$ is the half-activation voltage, F is Faraday's constant, R is the gas constant and T is temperature in Kelvin.

Time constants (τ) for nanobody binding and inactivation were obtained by fitting a single or double exponential function to the traces for depolarization between +20 mV to +50 mV. The traces were fitted from the beginning of the decaying phase to the end of the depolarizing trace using the following equation:

$$f(t) = \sum_{i=0}^{n} A_i e^{-t/\tau_i} + C \quad (2)$$

Sigmoidicity values for recovery from nanobody inhibition were obtained by fitting the following equation[70] to the data obtained from the time-course experiments (Fig. 4D):

$$I_K = A\left(1 - e^{-t/\tau_{off}}\right)^{\sigma} \quad (3)$$

The amplitude of the sigmoidal curve A develops with a time constant $\tau_{off}$ (s), and sigmoidicity σ, which is unitless. When σ = 1, the equation describes a monoexponential rise, as would be expected from a process involving one transition. As σ increases the value of σ would indicate the number of transitions required to produce such a sigmoidicity.

**Cryo-EM sample preparation and data acquisition**. Samples were prepared using human Kv1.3 protein (3 mg/mL) without a binding partner, or with nanobody at 1.3 mg/mL (3:1 molar ratio, nanobody:Kv1.3 subunit), or with Fab-ShK at 0.4 mg/mL (1:1 molar ratio, Fab-ShK:Kv1.3 tetramer). UltrAuFoil 1.2/1.3 300 mesh grids (Quantifoil) were plasma treated and vitrified samples were prepared by adding a 2.5 μL droplet of sample solution to a grid, then blotting (2 sec blot time, 0 to −4 blot force range) and plunge-freezing in liquid ethane using a Vitrobot Mk IV (Thermo Fisher).

Single particle images were collected with a Titan Krios electron microscope (Thermo Fisher) operated at 300 kV and a nominal magnification of 105,000× and equipped with a K3 camera (Gatan) set in super-resolution mode (0.4260 Å pixel size). Leginon was used for automated collection of images with ice thickness ranging between 20 nm and 120 nm[71,72]. Movies were collected at nominal defocus values of ~1.0–2.0 μm and dose-fractionation into 48 frames with a total exposure time of 2.4 sec and total dose of ~51 e-Å$^{-2}$.

**Image processing and structure analysis**. Movie stacks were corrected for beam-induced motion with two-fold binning and dose-weighted in Relion[54]. The resulting images were used for contrast transfer function (CTF) estimation with CTFFIND4.1[73]. Datasets were then processed as follows.

For the unbound Kv1.3 dataset an initial particle set was picked with the reference-free Laplacian-of-Gaussian (LoG) tool that is part of the Relion package, and particles were extracted with a box size of 320 and downscaled to 128. Particles were imported into cryoSPARC[74] and processed with 2D classification, ab initio reconstruction and heterogeneous refinement to generate a well-defined particle subset. The subset was re-extracted in Relion using a box size of 320 and the 3D model generation tool in Relion was used to create a template map for particle picking. Particle picking was done with an interparticle distance of 150 Å to avoid picking duplicates and yielded 526,276 particles. The new particle set was extracted with a box size of 320 and imported into cryoSPARC. Particles were processed with ab initio reconstruction and multiple rounds of heterogeneous refinement to isolate a subset of 190,793 particles which were refined with homogeneous and non-uniform refinement to 3.30 Å with C1 symmetry. Particle polishing was done in Relion and ab initio reconstruction, homogeneous refinement and non-uniform refinement were done in cryoSPARC to obtain a 3.12 Å map with C1 symmetry. A final non-uniform refinement with C4 symmetry was used to generate a map with 2.89 Å resolution.

The Kv1.3 dataset was further processed to isolate maps for the D1 and D2 subunit conformations. The particle set used to generate the C4 symmetric tetramer map (190,793 tetramer particles) was subjected to symmetry expansion in Relion to generate images of individual tetramer subunits (763,172 subunit particles). The subunit images were processed with multiple rounds of 3D classification without alignment in Relion using a mask around a single subunit. Uninterpretable classes or classes with a mixture of D1/D2 loop conformations were discarded, and D1 and D2 subunit maps were resolved with 107,046 and 217,344 particles, respectively. These particles were used for additional analysis on tetramer compositions as presented in Supplementary Table 2.

For the Kv1.3 with nanobody dataset an initial particle set was picked with the LoG tool in Relion and particles were extracted with a box size of 416 and downscaled to 128. Particles were imported into cryoSPARC and approximately half of the LoG-picked particles were randomly selected and processed with 2D classification, ab initio reconstruction and heterogeneous refinement to generate a well-defined subset of 27,427 particles. The subset was re-extracted in Relion using

a box size of 416 and the 3D model generation tool in Relion was used to create a template map for particle picking. Particle picking was done with an interparticle distance of 150 Å to avoid picking duplicates and yielded 394,303 particles. The new particle set was extracted with a box size of 416 and imported into cryoSPARC. Particles were processed with multiple rounds of heterogeneous refinement to isolate a subset of 138,728 particles which were refined with homogeneous and non-uniform refinement to 3.62 Å with C4 symmetry. Particle polishing was done in Relion and ab initio reconstruction and heterogeneous refinement were done in cryoSPARC to isolate a subset of 123,722 polished particles. These particles were further refined with homogeneous refinement and non-uniform refinement to generate a final map at 3.25 Å resolution map with C4 symmetry.

For the Kv1.3 with Fab-ShK dataset an initial particle set was picked with the LoG tool in Relion, extracted with a box size of 440 and downscaled to 128, and processed with 2D classification in cryoSPARC to identify an interpretable subset. Templates were generated with 2D classification in Relion and used for particle picking to yield 482,405 particles. The particles were extracted with a box size of 440, imported into cryoSPARC, and 2D classification was used to isolate a subset of 447,444. Particles were subjected to iterative heterogeneous refinement in cryoSPARC to isolate 90,267 particles that were used for non-uniform refinement to generate a 3.71 Å map. Kv1.3 and the ShK toxin were well-defined in the map, but only the overall molecule profile was visible for the Fab region. Attempts to improve the Fab region with local refinement were unsuccessful. The particles were subjected to particle polishing in Relion and then refined in cryoSPARC with non-uniform refinement to a final resolution of 3.39 Å. C1 symmetry was used throughout.

**Structural modeling**. A homology model for Kv1.3 was generated using PDB 2R9R (Kv1.2 crystal structure) (Kv1.2-Kv2.1 structure). The homology model was docked into each Kv1.3 map using Chimera[75] and the model for each structure refined using Coot[76]. For the unbound Kv1.3 structure, separate models were built for the D1 and D2 conformations. This was done by using the D1 or D2 subunit maps in parallel with the tetrameric C4 symmetric consensus map. The two models are virtually identical except for the selectivity filter region. For the Kv1.3 with nanobody A0194009G09 structure a homology model was generated using PDB 6V80 (NKT12 TCR and VHH nanobody 1D12 structure), docked into the map using Chimera, and refined using Coot. Separate versions of the Kv1.3-NB complex were modeled for the D1 and D3 conformations, which varied differed only in the rotamer orientation of Tyr447 and associated local changes in the protein backbone. For the map of Kv1.3 with the Fab-ShK fragment from MNT-002, a homology model was generated from PDB 4LFQ (ShK) (L-ShK toxin structure) and PDB 6OO0 (bovine Fab) (bovine Fab NC-Cow1 structure) and docked into the map using Chimera. The ShK toxin model was refined using Coot. The Fab region of the map was sufficient to orient the Fab but was not sufficient for further modeling so all residues were stubbed to alanine. The extracellular VSD linkers were not resolved so were not modeled. The exception to this is the first segment of the S1-S2 loop which was stabilized by nanobody binding in the Kv1.3-nanobody complex so could be modeled.

**Reporting summary**. Further information on research design is available in the Nature Research Reporting Summary linked to this article.

## Data availability

The cryo-EM density maps and models for Kv1.3, Kv1.3 with nanobody A0194009G09, and Kv1.3 with the Fab-ShK from MNT-002 have been deposited in the Protein Data Bank (PDB) and Electron Microscopy Data Bank and will be released upon publication. The PDB accession codes are PDB 7SSX (Kv1.3 D1), PDB 7SSY (Kv1.3 D2), PDB 7SSZ (Kv1.3 with nanobody D1), PDB 8DFL (Kv1.3 with nanobody D3) and PDB 7SSV (Kv1.3 with Fab-ShK) and the EMD accession codes are EMD-25416 (Kv1.3 map, D1 subunit map, D2 subunit map), EMD-25417 (Kv1.3 with nanobody) and EMD-25414 (Kv1.3 with Fab-ShK). Source data are provided with this paper.

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

## Acknowledgements

We thank Tsg-Hui Chang for frog keeping and surgery, Abigail Kelley for technical support, Francis I. Valiyaveetil for providing Kv1.2-2.1 coordinates, and members of our labs, Alessio Accardi's lab and Olga Boudker's lab for discussion. Cryo-EM was performed at the NYU Langone Health's Cryo-Electron Microscopy Laboratory, the National Cancer Institute's National Cryo-EM Facility (NCEF) at the Frederick National Laboratory for Cancer Research under contract HSSN261200800001E, and at the Simons Electron Microscopy Center and National Resource for Automated Molecular Microscopy located at the New York Structural Biology Center and supported by grants from the Simons Foundation (SF349247), NYSTAR, and the NIH National Institute of General Medical Sciences (GM103310). Assistance with cryo-EM was provided by Tara Fox and Thomas Edwards (NCEF), and Eugene Chua and Edward T. Eng (New York Structural Biology Center). Molecular graphics were prepared and analysis performed with UCSF ChimeraX[77], developed by the Resource for Biocomputing, Visualization, and Informatics at the University of California, San Francisco with support from National Institutes of Health R01-GM129325 and the Office of Cyber Infrastructure and Computational Biology, National Institute of Allergy and Infectious Diseases. This work was supported by NIH grant R01 GM105826 to V.V.S., National Institute of Neurological Disorders and Stroke NS002945 to K.J.S., and a Leon Levy Fellowship in Neuroscience grant and a STARR Cancer Consortium grant to J.R.M.

## Author contributions

P.S. purified Kv1.3 and the A0194009G09 nanobody; R.H. purified the MNT-002 Fab; P.S., A.J.P., B.W. and W.J.R. performed cryo-EM; P.S., N.K., C.H. and J.R.M performed structure determination; A.I.F.M. performed electrophysiological experiments; PS and JRM wrote the manuscript; P.S., A.I.F.M., N.K., C.H., A.J.P., B.W., R.H., V.V.S., W.J.R., K.J.S. and J.R.M. contributed to manuscript editing and preparation.

## Competing interests

V.V.S. and R.H. have an equity interest in Minotaur Therapeutics which has a license to the MNT-002 molecule. The authors declare no other competing interests.
