## [Peer Review File · Nature Communications]

Structures of the T cell potassium channel Kv1.3 with immunoglobulin modulatorsReviewer #1 (Remarks to the Author):

The manuscript by Selvakumar et al reports the structure of the voltage-gated potassium channel Kv1.3 in two, presumably, inactivated states at 2.89 Å, two states binding a Kv1.3 specific nanobody on the turret loops and a state blocked by a ShK-Fab construct. These are very important findings for the field because 1) Kv1.3 constitutes an attractive target for immunosuppression and the structures can enable the design of novel immunosuppressants and 2) the structures show novel, previously not captured selectivity filter conformations for Kv channels and thus provide novel insights into pore dynamics and the mechanisms of gating and permeation.

The manuscript is well written, the experiments beautifully conducted and the methods described in sufficient detail. The overall findings are very much in line with the biophysical properties of Kv1.3 and the ShK contacts are in agreement with previous findings. It, however, is a surprise how flexible the selectivity filter is and how many different, dilated, inactivated conformations exist. That is a tremendous new insight into the "life of Kv channels."

As a reviewer I only have two comments on this beautiful study.

1. The authors should comment in more detail on the resolution of the different structures. Is it the same all over the structures? Which portions of the channel are not resolved? The Apo structure shown in Figure 1 is obviously missing the S1-S2 and the S3-S4 linkers. However, this is never commented on. The nanobody bound structure seems to contain a resolved S3-S4 linker. This is a long, 27-residue linker. The authors should comment on the structure of this linker and whether the nanobody is making any contacts with this linker or whether there are only contacts to the turret loop and the S1-S2 linker (as shown in Figure 3).

2. The stoichiometry of the nanobody binding is obviously variable and dependent on the ratios of nanobody to channel protein. The authors are providing on and off rates at two different concentrations (10 nM and 100 nM) demonstrating different occupancies. It would be nice if the authors could supplement this data with a full concentration response curve for the nanobody and then comment on the IC50 and the Hill coefficient.

Reviewer #2 (Remarks to the Author):

In this timely report by Selvakumar and colleagues, the authors characterize the inhibition of Kv1.3 channels by a nanobody inhibitor and an engineered antibody toxin fusion inhibitor. This works nicely complements a recently published by Tyagi and colleagues, which characterized the inhibition of Kv1.3 by Dalazatide, a similar engineered antibody toxin fusion inhibitor (PMID: 35091471). In the current study, the authors first describe structures of Kv1.3 in ligand-free states. Two distinct conformations of the selectivity filter are described that differ from the selectivity filters resolved in structures of related Kv1.2 channels, which the authors attribute to the channels undergoing C-type inactivation. In the presence of the nanobody inhibitor, which binds to the peripheral VSDs, the selectivity filter adopts two distorted states. One of these closely resembles one of the apo distorted states and the second is distinct from the apo states. The engineered antibody toxin fusion inhibitor, which acts as a pore blocker, binds to the ion selectivity filter, and occludes ion permeation. Its binding, largely through a lysine residue that extends into the selectivity filter, stabilizes a conductive state, similar to the conductive state resolved by Tyagi and colleagues for Kv1.3 bound to Dalazatide. The structural studies are nicely supported by electrophysiological analyses and together provide a detailed analysis of Kv1.3 inhibition. However, the text is quite brief and many of the findings are inadequately described. For example, introduction is too brief and lacks a sufficient introduction to potassium channel selectivity filters and C-type inactivation to help non-experts appreciate the findings. In addition to expanding the text, I have some suggestions below that would strengthen the analysis of these excellent data. In a revised version with these comments addressed, this report would be suitable for publication in Nature Communications.

Comments.

1. In Figure 1, the authors highlight the caveolin binding domain of each protomer. However, these features are insufficiently described. Why were they highlighted? Are they unique to Kv1.3 and are there any known roles? Do the structures inform on these domains' functions in a meaningful way?
2. It is hard to appreciate the distortions of the selectivity filter. There are no dimensions shown, nor is there an overlay of the various states. For example, how close are the opposing atoms in the selectivity filter? Additional details to the existing panels or the introduction of additional panels would be helpful in comparing the states.
3. It would also be helpful if the authors present the densities for the ions in both C1 and C4 in the main text as the selectivity filter appears to be highly distorted and it would be helpful to visualize the effects of these distortions on the relative ion occupancies.
4. The authors attempt to quantify the relative abundance of the two dilated conformations. However, as only a small fraction of the protomers imaged could be classified (~325k/763k), it is not clear that the quantification is representative. Similarly, the classification presented in Table S2 only for the relative abundance of the states within tetramers corresponds to only ~7k/190k tetramers classified. These authors need to clarify that their analysis only corresponds to a fraction of the data and that these particles may not be representative.
5. It is difficult to discern the binding interface presented in Figures 3I-J and 4D-E because the side chains are all colored similarly, as are the labels. Again, it would be helpful to show dimensions and an overlay for the various selectivity filter conformations in figures 3 and 4.
6. The authors describe numerous hydrogen bonds in the text and the figures. However, the location and distances of these bonds are not shown in the figures. In many cases the figures are too crowded and small to see the details. For example, in Figure 4F, the legend describes the panel as "Key hydrogen bonds involved in channel gating are stabilized when Kv1.3 is bound by ShK", yet it is not clear which residues are hydrogen bonded. It would be helpful if the authors increased the size of several of figures to add in additional details.

Reviewer #3 (Remarks to the Author):

Kv1.3 is expressed on T lymphocytes and is upregulated during activation of the immune response. It is essential for immune function. If this channel is inhibited or unable to conduct K⁺ ions, immune suppression occurs, a key therapeutic strategy to treat the vast array of autoimmune diseases. To address this challenge, the authors obtain cryo-EM structures of Kv1.3 in complex with immunoglobulin-based Kv1.3 modulators. They demonstrate two distinct mechanisms to suppress Kv1.3 channel activity. One mechanism involves nanobody binding to induce an inactive pore conformation, putatively a C-type inactivated state. The other involves docking of an antibody-toxin fusion molecule into the extracellular mouth of the channel to block the K⁺ permeation pathway. Dynamics between pore conformations, including rotamers, H-bond interactions, and hydrophobic interactions suggest potential mechanisms for immunotherapeutic development. This is clearly an impactful paper. However, comments and criticisms, as outlined below, must be addressed.

General Comments

- 1). The structures appear to be obtained in 150mM K⁺, but the electrophysiology is done at ~ 2 mM K⁺. The latter conditions support "C-type" inactivation; the former inhibit it (and also speed recovery). If so, then what states do the D1 and D2 structures represent? Are they physiologically relevant and reflect states manifest in the physiological range of extracellular K⁺ concentrations? The gating state is ion-dependent. Moreover, K⁺ is a structural cofactor of K⁺ channels and they are not "stable" with low K⁺, perhaps requiring many more averaging events. Kv1.3 has a low affinity K⁺ binding site that modulates recovery from slow inactivation, likely conformationally dependent and implicating a role in structural stability and a physiological role for autoregulation of Kv1.3 by extracellular K⁺. The critical experiment is not shown and should be done, i.e., the proper control for comparison with the cryo-EM structure to identify its gating state is to record for a long time at 0 mV at extracellular 150 mM K⁺ (the cryo-EM conditions). Other hallmarks of C-type inactivation exist but are not invoked in cryo-EM samples to verify the assigned gating status of D2. They should be done and reported.

2). The 100 mM nanobody-treated channels do not totally inactivate at +50/30mV (Fig. 3A). Could there be a minor population of non-inactivating current? Would this be expected if nanobody induces increased dynamics between active and inactive conformations under the electrophysiological conditions? Even if there is 5% or 10% non-inactivating current, that will complicate the interpretation. Perhaps those non-inactivating channels are more amenable for the structure work. For a Shaker W434F and another mutant, there is no ionic current at low K⁺. So once you see non-inactivating current, how does this influence the interpretations?

3). Please clarify the use of the C4 symmetry model constraints. Some tetramers contain a mixture of D1 and D2 subunits (D1:D2 1:3, 30.8%; D1:D2 2:2, 24.5%) and the majority of tetramers are a mixture of D1 and D2 subunits ("suggesting that the D1 and D2 filter conformational changes do not occur in a fully cooperative manner"). How can the C4 symmetry algorithm be applied? For the control structure, the symmetry constraint may be ok, but once you start adding agonist-bound moieties, what are the real constraints? Please clarify. At the end, the dilation is presented as a symmetrical dilation yielding "Four copies of NBs are observed." If the symmetry assumption is used, aren't you then guaranteed to see 4 copies?

4). The authors state that D2 resembles structures of mutant Shaker and chimera, which are proposed to be C-type inactivated conformations. This is not compelling and needs more definitive demonstration. The cited "precedents" may have similar conceptual caveats (due to 150 mM vs low K⁺, 0 mV vs ≥+50 mV; see Comment above).

5). The pore helix in Kv1.3 structures (D1 & D2) in Figures 2-4 appear conformationally distinct from the pore helix in the chimera (Kv1.2/2.1, Fig. 2D, G, J), albeit this is from cartoon representations. What does a rigorous pore helix comparison show? They might be expected to be different if conditions were 150 mM K⁺ for the Kv1.3 versus a different [K⁺] for the chimera. Please clarify and interpret.

6). Why was ShK fused to the Fab fragment since ShK by itself blocks the ion permeation pathway with high affinity? The K_d of ShK is 11 pM. What is ShK's off rate? What is Fab-ShK's off rate? Please state. Nonetheless, the authors need to explain the motivation behind making the fusion and say more about Shk. Furthermore, the authors neither show nor mention any direct interactions between the Fab fragment and Kv1.3; Fig. 4 shows Fab hovering out in space on a tether to ShK. So why make the fusion complex? Could Fab's bulk decrease the ShK on/off rates? What are they? I suspect a biological/pharmaceutical purpose motivated making the fusion complex. This should be explicitly articulated. One sentence in the Ms hints at this "The Fc region of the antibody engages Fc receptors to enhance inhibition of T cells (manuscript in preparation)". Please elaborate if this is the reason.

The final conclusions are reasonable, but there are many loose ends in the manuscript, which need to be explained and additional data supplied.

Minor Comments

- 1). Add a schematic or structural image for ShK and for nanobody to Fig. 1A, so the reader is not just being introduced to an acronym. Give MW, K_d etc for ShK to inform the reader. Alternatively, for ShK and Fab-ShK, add this information to Fig.4A.
- 2). Fig. S5D could be redrawn so symbols are easier to see. Fig. S5E and its legend are a little confusing. Please redo both to clarify for the reader. Specify [K⁺] for A and C in the legend.
- 3). At the end of the Results, when you state "...Ser20 and Arg11 are major determinants of ShK toxin binding" presumably via the interaction with His 451, does this have a lower k_{off} and higher affinity in comparison to other channels or His mutations? In these cases, what happens to the K_d and the off rate of ShK? Do the other Kv1 isoform residues make weaker H-bonds to ShK Ser20/Arg 11?

REVIEWER COMMENTS

Reviewer #1 (Remarks to the Author):

The manuscript by Selvakumar et al reports the structure of the voltage-gated potassium channel Kv1.3 in two, presumably, inactivated states at 2.89 Å, two states binding a Kv1.3 specific nanobody on the turret loops and a state blocked by a ShK-Fab construct. These are very important findings for the field because 1) Kv1.3 constitutes an attractive target for immunosuppression and the structures can enable the design of novel immunosuppressants and 2) the structures show novel, previously not captured selectivity filter conformations for Kv channels and thus provide novel insights into pore dynamics and the mechanisms of gating and permeation.

The manuscript is well written, the experiments beautifully conducted and the methods described in sufficient detail. The overall findings are very much in line with with the biophysical properties of Kv1.3 and the ShK contacts are in agreement with previous findings. It, however, is a surprise how flexible the selectivity filter is and how many different, dilated, inactivated conformations exist. That is a tremendous new insight into the "life of Kv channels."

As a reviewer I only have two comments on this beautiful study.

1. The authors should comment in more detail on the resolution of the different structures. Is it the same all over the structures?

We thank the reviewer for the comments and this is an important point. The resolution does indeed vary throughout the structures. This variation is common with cryo-EM structures and the standard technique to visualize variable resolution is with a local resolution heat map. These heat maps were calculated for our structures and are found in Figure S1G (Kv1.3 unbound), Figure S3E (Kv1.3 with nanobody), and Figure S5D (Kv1.3 with Fab-ShK).

For all structures the heat maps show the general trend that resolution is highest in the pore region, ranging from ~2-3 Å for all structures. In all maps the resolution is slightly attenuated in the VSDs and ranges from ~3-3.5 Å, and in the T1 domains is in the range of 3-4.5 Å.

For the nanobody bound structure, the nanobodies are well-resolved near the interface with Kv1.3 (~3 Å) and the nanobody resolution progressively diminishes in the regions distal to the channel (~3.5-4 Å).

For the Fab-ShK bound structure, the ShK domain has resolution of ~3 Å while the Fab is not well-visualized. We speculate that the attenuated structural features in the Fab are due to a degree of flexibility in the "stalk" region which connects ShK to the Fab.

In addition to the heat maps, we also presented isolated cryo-EM densities for the major helices in the unbound Kv1.3 structure to convey the quality of the cryo-EM (Figure S1II).

We have added text in the manuscript to clarify that resolution varies throughout the structures (Lines 101-105; Lines 192-198; Lines 249-254)

Which portions of the channel are not resolved?

In all the structures the following regions are not resolved: S1-S2 linkers, S3-S4 linkers, unstructured N-terminus (102 residues), and the unstructured C-terminus (84 residues). The exception is an additional 9 residues of the S1-S2 linkers (D261-T269) which are resolved in the structure of Kv1.3 with the NB due to linker interactions with the nanobody.

This point is now clarified in the text in the lines noted above.

The Apo structure shown in Figure 1 is obviously missing the S1-S2 and the S3-S4 linkers. However, this is never commented on. The nanobody bound structure seems to contain a resolved S3-S4 linker. This is a long, 27-residue linker. The authors should comment on the structure of this linker and whether the nanobody is making any contacts with this linker or whether there are only contacts to the turret loop and the S1-S2 linker (as shown in Figure 3).

Indeed, the Apo structure is missing the S1-S2 and S3-S4 linkers. Although the nanobody contacts the S1-S2 linker, the S3-S4 linker is missing in the nanobody-bound structure so we concluded that the nanobody does not make direct contact with S3-S4.

We have added text to make this conclusion clear to the reader in the lines noted above.

2. The stoichiometry of the nanobody binding is obviously variable and dependent on the ratios of nanobody to channel protein. The authors are providing on and off rates at two different concentrations (10 nM and 100 nM) demonstrating different occupancies. It would be nice if the the authors could supplement this data with a full concentration response curve for the nanobody and then comment on the IC50 and the Hill coefficient.

We agree with the reviewer that it would be valuable to see other concentrations in addition to 10 nM and 100 nM, and we have now tested the nanobody at 1 nM and 1 uM and the data is presented in Fig. 4. However, we have not used this data to evaluate the stoichiometry of NB interaction with Kv1.3 because the exceedingly slow off-rate of the nanobody makes it challenging to be confident that we have achieved equilibrium at the lower concentration (1 or 10 nM).

Reviewer #2 (Remarks to the Author):

In this timely report by Selvakumar and colleagues, the authors characterize the inhibition of Kv1.3 channels by a nanobody inhibitor and an engineered antibody toxin fusion inhibitor. This works nicely complements a recently published by Tyagi and colleagues, which characterized the inhibition of Kv1.3 by Dalazatide, a similar engineered antibody toxin fusion inhibitor (PMID: 35091471). In the current study, the authors first describe structures of Kv1.3 in ligand-free states. Two distinct conformations of the selectivity filter are described that differ from the selectivity filters resolved in structures of related Kv1.2 channels, which the authors attribute to the channels undergoing C-type inactivation. In the presence of the nanobody inhibitor, which binds to the peripheral VSDs, the selectivity filter adopts two distorted states. One of these closely resembles one of the apo distorted states and the second is distinct from the apo states. The engineered

antibody toxin fusion inhibitor, which acts as a pore blocker, binds to the ion selectivity filter, and occludes ion permeation. Its binding, largely through a lysine residue that extends into the selectivity filter, stabilizes a conductive state, similar to the conductive state resolved by Tyagi and colleagues for Kv1.3 bound to Dalazatide. The structural studies are nicely supported by electrophysiological analyses and together provide a detailed analysis of Kv1.3 inhibition. However, the text is quite brief and many of the findings are inadequately described. For example, introduction is too brief and lacks a sufficient introduction to potassium channel selectivity filters and C-type inactivation to help non-experts appreciate the findings. In addition to expanding the text, I have some suggestions below that would strengthen the analysis of these excellent data. In a revised version with these comments addressed, this report would be suitable for publication in Nature Communications.

We thank the reviewer for the feedback and we have expanded the introduction to include more background on potassium channel selectivity and C-type inactivation (Introduction section, paragraphs 2 and 3). We have also used the reviewer's other suggestions to guide additions and improve presentation and clarity for the reader. These are detailed below.

Comments.

1. In Figure 1, the authors highlight the caveolin binding domain of each protomer. However, these features are insufficiently described. Why were they highlighted? Are they unique to Kv1.3 and are there any known roles? Do the structures inform on these domains' functions in a meaningful way?

It was recently discovered that Kv1.3 harbors a caveolin binding domain which is responsible for Kv1.3 localization to the plasma membrane (Capera *et al.* 2021 *eLife*). To our knowledge this is a unique attribute of Kv1.3 compared to other Kv1 channels. We highlighted the domain to draw attention to this physiologically important structural feature.

However, upon consideration we feel that this aspect of Kv1.3 is not relevant to our study. To maintain the focus and clarity of the manuscript we decided to remove the highlight. This change is reflected in the revised version of Figure 1.

2. It is hard to appreciate the distortions of the selectivity filter. There are no dimensions shown, nor is there an overlay of the various states. For example, how close are the opposing atoms in the selectivity filter? Additional details to the existing panels or the introduction of additional panels would be helpful in comparing the states.

This is a good point and we agree with the reviewer's assessment. We have added additional panels, scale bars and distance measurements to all main figures (except for Figure 4 which does not include any structural analysis). Among these additions we now present distance measurements between opposing atoms in the selectivity filter. Finally, we have added a new supplemental figure (Figure S7) to show structural overlays of various states.

3. It would also be helpful if the authors present the densities for the ions in both C1 and C4 in the main text as the selectivity filter appears to be highly distorted and it would be helpful to visualize the effects of these distortions on the relative ion occupancies.

We have added panels in Figure 1 to show the ion densities in both the C1 and C4 maps. We concluded from maps that the ion occupancies are roughly equal, and have now noted this in the main text (Lines 136-137).

0. The authors attempt to quantify the relative abundance of the two dilated conformations. However, as only a small fraction of the protomers imaged could be classified (~325k/763k), it is not clear that the quantification is representative. Similarly, the classification presented in Table S2 only for the relative abundance of the states within tetramers corresponds to only ~7k/190k tetramers classified. These authors need to clarify that their analysis only corresponds to a fraction of the data and that these particles may not be representative.

We thank the reviewer for this comment and agree it is important to note that the subset analyzed may not be representative. We have added text to clearly present this point to the reader (Lines 173-177).

1. It is difficult to discern the binding interface presented in Figures 3I-J and 4D-E because the side chains are all colored similarly, as are the labels. Again, it would be helpful to show dimensions and an overlay for the various selectivity filter conformations in figures 3 and 4.

To improve legibility in Figures 3I-J (now Figures 3K-L), we have assigned different colors to the nanobody and Kv1.3 sidechains. For Figure 4D-E (now Figures 5F-G) we have recolored the side chains of the ShK molecule to clearly distinguish them from the Kv1.3 sidechains.

We have added dimensions throughout Figures 3 and 4 (now Figures 3 and 5), and labels on the selectivity filter depictions in both figures (Figure 3J and 5H). Structural alignments between different structural states are provided in Figure S7,

2. The authors describe numerous hydrogen bonds in the text and the figures. However, the location and distances of these bonds are not shown in the figures. In many cases the figures are too crowded and small to see the details. For example, in Figure 4F, the legend describes the panel as “Key hydrogen bonds involved in channel gating are stabilized when Kv1.3 is bound by ShK”, yet it is not clear which residues are hydrogen bonded. It would be helpful if the authors increased the size of several of figures to add in additional details.

We acknowledge this presentation issue in the original manuscript and to fix it we have indicated the hydrogen bonds and added distance measurements in all relevant figure panels. Figure panel sizes have also been increased to improve legibility.

Reviewer #3 (Remarks to the Author):

Kv1.3 is expressed on T lymphocytes and is upregulated during activation of the immune response. It is essential for immune function. If this channel is inhibited or unable to conduct K⁺ ions, immune suppression occurs, a key therapeutic strategy to treat the vast array of autoimmune diseases. To address this challenge, the authors obtain cryo-EM structures of Kv1.3 in complex with immunoglobulin-based Kv1.3 modulators. They demonstrate two distinct mechanisms to

suppress Kv1.3 channel activity. One mechanism involves nanobody binding to induce an inactive pore conformation, putatively a C-type inactivated state. The other involves docking of an antibody-toxin fusion molecule into the extracellular mouth of the channel to block the K⁺ permeation pathway. Dynamics between pore conformations, including rotamers, H-bond interactions, and hydrophobic interactions suggest potential mechanisms for immunotherapeutic development. This is clearly an impactful paper. However, comments and criticisms, as outlined below, must be addressed.

General Comments

1). The structures appear to be obtained in 150mM K⁺, but the electrophysiology is done at ~ 2 mM K⁺. The latter conditions support “C-type” inactivation; the former inhibit it (and also speed recovery). If so, then what states do the D1 and D2 structures represent? Are they physiologically relevant and reflect states manifest in the physiological range of extracellular K⁺ concentrations? The gating state is ion-dependent. Moreover, K⁺ is a structural cofactor of K⁺ channels and they are not “stable” with low K⁺, perhaps requiring many more averaging events. Kv1.3 has a low affinity K⁺ binding site that modulates recovery from slow inactivation, likely conformationally dependent and implicating a role in structural stability and a physiological role for autoregulation of Kv1.3 by extracellular K⁺. The critical experiment is not shown and should be done, i.e., the proper control for comparison with the cryo-EM structure to identify its gating state is to record for a long time at 0 mV at extracellular 150 mM K⁺ (the cryo-EM conditions).

These are excellent points. We have performed new experiments (presented in Fig S2) where we measure the time course and extent of steady-state inactivation using 20 sec step depolarizations at low and high extracellular K⁺ concentrations (2 mM and 100 mM) using two-electrode voltage clamp. Because oocytes contain 100 mM internal K⁺, the highest concentration of external K⁺ we could test is 100 mM. Although these data show that increasing concentrations of external K⁺ slow the entry of channels into inactivated states and diminishes the fraction of inactivated channels at 0 mV, most channels remain inactivated at 100 mM external K⁺ and at 0 mV, conditions similar to those used for structure determination. Therefore, we think the D2 conformation we observed is likely to represent an inactivated state principally because it exhibits reduced ion coordination compared to Kv channel structures that are thought to represent conductive states, and it shows rupture of two hydrogen bonds which are known to control entry into the C-type inactivated state (Pless *et al.* 2013 *eLife*). In addition, the structure is similar to that reported for C-type inactivated mutants of a Shaker Kv channel and a Kv1.2-2.1 channel (Tan *et al.* 2022 *Sci. Adv.*; Reddi *et al.* 2021 *bioRxiv*). We think the D1 state may be an intermediate between the inactivated state and the conducting state. Importantly, we note our ability to restore a conformationally homogeneous, active-like conformation of the channel by adding ShK. We have added text to clarify that the D1 and D2 conformations may represent intermediate and inactivated states of the channel (Lines 153161).

Other hallmarks of C-type inactivation exist but are not invoked in cryo-EM samples to verify the assigned gating status of D2. They should be done and reported.

We apologize in advance, but we do not understand what the reviewer is referring to in this statement.

2). The 100 mM nanobody-treated channels do not totally inactivate at +50/30mV (Fig. 3A). Could there be a minor population of non-inactivating current? Would this be expected if nanobody induces increased dynamics between active and inactive conformations under the electrophysiological conditions? Even if there is 5% or 10% non-inactivating current, that will complicate the interpretation. Perhaps those non-inactivating channels are more amenable for the structure work. For a Shaker W434F and another mutant, there is no ionic current at low K⁺. So once you see non-inactivating current, how does this influence the interpretations?

We now provide data at higher NB concentrations and at elevated K⁺ (Fig. 3 and Fig. S2). Although we cannot exclude the possibility of a small fraction of non-inactivated channels at 0 mV and high K⁺, our results show that a majority of the channels are inactivated under those conditions. On the grid with a 3 molar excess of nanobody to channel (far higher than we used in the functional experiments) we see four nanobodies bound to each channel. Taking into consideration the functional results discussed above, we expect that the channel on the grids would be largely inactivated regardless of the high K⁺ concentration used.

3). Please clarify the use of the C4 symmetry model constraints. Some tetramers contain a mixture of D1 and D2 subunits (D1:D2 1:3, 30.8%; D1:D2 2:2, 24.5%) and the majority of tetramers are a mixture of D1 and D2 subunits ("suggesting that the D1 and D2 filter conformational changes do not occur in a fully cooperative manner"). How can the C4 symmetry algorithm be applied?

This is an important point and the logic of using C4 model constraints is as follows. Aspects of this response are in the manuscript but are briefly presented again here just for clarity.

During image processing for the unbound Kv1.3 structure we observed that all four subunits showed a mixture of two pore conformations which we named D1 and D2. In addition, the two conformations were observed both without symmetry (C1) or with symmetry (C4) applied to the data. To separate the conformations we extracted individual subunits and resolved 3D cryo-EM maps for each of the two subunit conformations (presented in Figure 2C). As part of this analysis we also derived statistics indicating that tetramers are a mixture of the subunits (as noted above by the reviewer).

Next, we aimed to model a tetrameric form of unbound Kv1.3. However, this was problematic because no single tetramer model could faithfully represent the conformational heterogeneity in the structure. As a compromise, we decided to simply model "pure" D1 and D2 tetramers (i.e. tetramers with C4 symmetry).

We feel that using C4 symmetric D1 and D2 tetramer models is the most practical way to present and disseminate the results. However, upon consideration we realize that this decision could lead to confusion. As such, we have added text in the manuscript to clarify this decision to readers (Lines 126-130).

For the control structure, the symmetry constraint may be ok, but once you start adding agonist-bound moieties, what are the real constraints? Please clarify. At the end, the dilation is presented as a symmetrical dilation yielding "Four copies of NBs are observed." If the symmetry assumption is used, aren't you then guaranteed to see 4 copies?

We initially processed the Kv1.3-nanobody dataset with C1 symmetry and observed four copies of the nanobody bound to Kv1.3. This supported the use of C4 symmetry for further processing of the data.

We have added text to clarify this important detail for the reader (Lines 189-192) and added panels to show the C1 symmetric map (Fig. S3A).

4). The authors state that D2 resembles structures of mutant Shaker and chimera, which are proposed to be C-type inactivated conformations. This is not compelling and needs more definitive demonstration. The cited “precedents” may have similar conceptual caveats (due to 150 mM vs low K⁺, 0 mV vs ≥+50 mV; see Comment above).

To demonstrate the similarity between D2, the Shaker mutant, and the Kv1.2-2.1 chimera mutant we have added a new supplemental figure with structural overlays (Fig. S7). The additional electrophysiological data from higher K⁺ condition discussed above further addresses this point.

5). The pore helix in Kv1.3 structures (D1 & D2) in Figures 2-4 appear conformationally distinct from the pore helix in the chimera (Kv1.2/2.1, Fig. 2D, G, J), albeit this is from cartoon representations. What does a rigorous pore helix comparison show? They might be expected to be different if conditions were 150 mM K⁺ for the Kv1.3 versus a different [K⁺] for the chimera. Please clarify and interpret.

To address the reviewer comment about K⁺ concentration: the Kv1.2-2.1 chimera structure (Long *et al.* 2007 *Nature*) and our Kv1.3 structures were solved in 150 mM KCl.

To enable a more rigorous pore comparison between D1, D2, and Kv1.2-2.1 we added labels which show the distances between backbone carbonyls in the selectivity filter along the length of the TVGYG motif (Fig. 2G-I, 3J, 5H). In addition we have added a new supplement (Fig. S7) which shows structural alignments of the pore structures analyzed in the manuscript. Our interpretations are below.

We interpret the measurements for Kv1.2-2.1 and Kv1.3 D1 and D2 as follows. The measurements show that bottom of the selectivity filter (TVGYG) is similar across the three structures, with a pore diameter of 4.7-5.1 Å. At the glycine in the middle of the filter (TVGYG) the Kv1.2-2.1 structure has a pore diameter of 4.7 Å while D1 and D2 are dilated to 6.6 Å and 9.3 Å respectively. The divergence between Kv1.2-2.1 and the Kv1.3 structures increases dramatically at the final two residues on the filter (TVGYG). Whereas the Kv1.2-2.1 pore has a maximum diameter of 9.9 Å, the Kv1.3 D1 and D2 conformations dilate to maximums of 14 Å and 11.6 Å. These measurements support the conclusion that D1 and D2 are conformationally distinct from each other, and from Kv1.2-2.1.

Next we consider the D1 and D3 conformations from Kv1.3 bound to the nanobody. The nanobody-bound D1 structure is similar to the D1 structure for unbound Kv1.3. For both D1 selectivity filters the distance between backbone carbonyls is ~5 Å at the base of the filter (TVGYG) and gradually tapers to a maximum distance of 14 Å (unbound D1) and 16.6 Å

(nanobody-bound D1) at the extracellular side of the filter. The relatively small numerical differences in the D1 selectivity filters, and visual inspection of the two structures, suggests they are qualitatively similar. This supports the decision to designate them both as D1. When comparing the D1 and D3 conformation of the nanobody-bound structure, we measured both small and large differences along the length of the selectivity filter (Fig. 3J). The most striking difference between the selectivity filters is the position of Tyr447 as described in the Results section, and depicted in Fig. 3I.

Pore measurements of the Kv1.3 complex with ShK further reveal its similarity to the Kv1.2-2.1 structure. Throughout the selectivity filter (TVGYG) the backbone carbonyl positions for the two structures differ by no greater than 0.3 Å (see Fig. 2G and Fig. 5H). The similarity between these structures is further visualized in Fig. S7A.

6). Why was ShK fused to the Fab fragment since ShK by itself blocks the ion permeation pathway with high affinity? The K_d of ShK is 11 pM. What is ShK's off rate? What is Fab-ShK's off rate? Please state. Nonetheless, the authors need to explain the motivation behind making the fusion and say more about Shk.

ShK was fused to the Fab regions of an IgG antibody (Figure 5A). The rationale was that ShK can target Kv1.3 and suppress T cell activation, while the Fc portion of the antibody could engage Fc receptors and send additional immune inhibitory signals. In addition, because IgG antibodies are divalent the antibody-ShK fusion provides the potential for increased avidity in blocking Kv1.3. It is also expected that the fusion will have enhanced half-life compared to ShK alone. We acknowledge that the physiological properties of the antibody are the subject of ongoing investigation. We have added these points to the manuscript to clarify the motivation for making the molecule (Lines 90-94).

The reason to generate the Fab-ShK was entirely technical. Using the divalent IgG-ShK would crosslink two copies of Kv1.3 together into a potentially flexible and unwieldy assembly, thus hindering cryo-EM image processing. In contrast, Fab-ShK is monovalent so can only bind one Kv1.3, thus forming a more rigid assembly amenable to cryo-EM. This is now explained in the text (Lines 240-243).

We have added data at different Fab-ShK concentrations to Figure 5 and we compare ShK and Fab-ShK in Fig. S4. We find that Fab-ShK and ShK exhibit similar activities. Although dissociation of both version of the toxin are extremely slow (Fig. S4D), they are also comparable.

Furthermore, the authors neither show nor mention any direct interactions between the Fab fragment and Kv1.3;

The Fab fragment does not make direct contact with Kv1.3. To convey this point we originally wrote that the Fab's "long and narrow CDR3 stalk allows it to avoid contact with the extracellular loops of Kv1.3" but we have now emphasized this point in the text (Lines 260-261).

Fig. 4 shows Fab hovering out in space on a tether to ShK. So why make the fusion complex? Could Fab's bulk decrease the ShK on/off rates? What are they?

The reason for using the Fab-ShK fragment is technical (see above for explanation).

Because the Fab region makes no direct contact with Kv1.3, we anticipated that it is unlikely to alter the on/off rates of the Fab-ShK fusion. Indeed, the new data added to Fig. S4 shows that the kinetics for the onset and recovery from inhibition by Fab-ShK and ShK are similar and that both Fab-Shk (Fig. 5) and Shk (Fig. S4) completely inhibit Kv1.3 at 10 nM.

I suspect a biological/pharmaceutical purpose motivated making the fusion complex. This should be explicitly articulated. One sentence in the Ms hints at this “The Fc region of the antibody engages Fc receptors to enhance inhibition of T cells (manuscript in preparation)”. Please elaborate if this is the reason.

We have elaborated on the motivation for making the IgG-ShK and Fab-ShK complexes in the explanation above, and it is now more explicit in the manuscript text.

The final conclusions are reasonable, but there are many loose ends in the manuscript, which need to be explained and additional data supplied.

Minor Comments

1). Add a schematic or structural image for ShK and for nanobody to Fig. 1A, so the reader is not just being introduced to an acronym. Give MW, Kd etc for ShK to inform the reader. Alternatively, for ShK and Fab-ShK, add this information to Fig.4A.

We have added a schematic for the nanobody and Fab-ShK in Figures 3A and 5A, respectively. The schematics also show the MW of the molecules.

2). Fig. S5D could be redrawn so symbols are easier to see. Fig. S5E and its legend are a little confusing. Please redo both to clarify for the reader. Specify [K+] for A and C in the legend.

We have changed the symbols to improve clarity. Figure S5 is now main Figure 4.

3). At the end of the Results, when you state “...Ser20 and Arg11 are major determinants of ShK toxin binding” presumably via the interaction with His 451, does this have a lower koff and higher affinity in comparison to other channels or His mutations? In these cases, what happens to the Kd and the off rate of ShK? Do the other Kv1 isoform residues make weaker H-bonds to ShK Ser20/Arg 11?

There are some data in the literature that are relevant to these points, most notably a study by Kalman *et al.* 1998 *JBC* where the authors investigated ShK binding to Kv1.1-Kv1.7 and reported IC50 values for each. Addressing the nature of toxin interactions with the different Kv1 family members and His mutants is an important topic of investigation. However, we feel the experiments and possibly mutant cryo-EM structures for that investigation would be beyond the scope of our current study.

REVIEWERS' COMMENTS

Reviewer #1 (Remarks to the Author):

The authors have addressed all of my previous concerns and clarified all questions I had. I am therefore simply restating what I previously said:

The manuscript by Selvakumar et al reports the structure of the voltage-gated potassium channel Kv1.3 in two, presumably, inactivated states at 2.89 Å, two states binding a Kv1.3 specific nanobody on the turret loops and a state blocked by a ShK-Fab construct. These are very important findings for the field because 1) Kv1.3 constitutes an attractive target for immunosuppression and the structures can enable the design of novel immunosuppressants and 2) the structures show novel, previously not captured selectivity filter conformations for Kv channels and thus provide novel insights into pore dynamics and the mechanisms of gating and permeation.

The manuscript is well written, the experiments beautifully conducted and the methods described in sufficient detail. The overall findings are very much in line with with the biophysical properties of Kv1.3 and the ShK contacts are in agreement with previous findings. It, however, is a surprise how flexible the selectivity filter is and how many different, dilated, inactivated conformations exist. That is a tremendous new insight into the "life of Kv channels."

Reviewer #2 (Remarks to the Author):

The authors have substantially improved the manuscript. If the authors can address my two remaining comments, the manuscript will be suitable for publication in Nature Communications.

Line 270. The authors claim that Fab-ShK binding induces an active conformation of the selectivity filter. Is it also possible that the channel can sample the active conformation, even if transiently, and that binding changes the equilibrium between the active and distorted configurations? The authors should temper their conclusion in the text for the title for figure 5.

Line 102. It would be better to avoid using an explicit resolution range for the reconstruction. While the plot suggests a maximum resolution of 2 Å, there are no features in the map consistent with a 2 Å reconstruction. Instead, state that the core of the protein is more well resolved compared to the periphery and especially to the T1 domains. Similar changes should be made when describing the local resolution of the other maps.

Reviewer #3 (Remarks to the Author):

The authors have made substantial revisions to the manuscript, which should now be accepted. Brief comments are as follows.

1). General Comments:

a) The authors have addressed the issue of 150mM vs 2 mM K⁺ and their known effects on C-type inactivation. These are presented in Fig. S2. "Although these data show increasing concentrations of external K⁺ slow the entry...and diminishes the fraction of inactivated channels at 0 mV, most channels remain inactivated at 100mM external K⁺ and 0 mV..." Yes, these added conditions are similar to those used for the cryo-EM determinations. This allows them to say "D2 conformation... is likely to represent an inactivate state..." Further justification is given by alluding to structure similarities to other published inactivated Kv channels. The statement that D1 "may be an intermediate between the inactivated state and the conducting state" is speculation, but all right for now.

2). Addressed.

3). The question regarding use of C4 model when the majority of tetramers are a mixture of D1 and D2 subunits (D1:D2 1:3, 30.8%; D1:D2 2:2, 24.5%) was addressed, although not wholly satisfactorily. As the authors point out, no single tetramer model is able to faithfully represent the conformational heterogeneity in the structure, and they offer a practical way to present the results, fully acknowledging the problem in lines 126-130. Good answer to symmetry constraints (lines 189-192 & C1 symmetry map addition to Fig. S3A).

4). Good response to issue of D2 semblance to mutant Shaker and chimera with addition of Fig. S7 as well as the newly added electrophysiological data at high K⁺.

5). All the new labels and distances added to Figs. 2 & 3 are enormously helpful and allow specific comparisons. Additions to Figs 2G-I, 3J, 5H, and the new Fig. S7 address the request for a more rigorous comparison of pores. These details give a more substantial and quantitative description of the structural differences between chimera, D1, and D2, which ultimately is necessary for further mechanistic interpretations going forward. Likewise, the detailed comparison D1 and D3 is informative.

It would be good if some of the discussion in the rebuttal could be included in the Ms, but I understand if the authors assume the reader can infer these conclusions from the added labels/inserted distances and prefer not to include a summarizing paragraph.

6). Excellent response. Author texts on Lines 90-94 and Lines 240-243 clarify the rationale for engineering/using Fab-ShK fusion and the technical reasons (obviates obstacles to cryo-EM image processing due to properties of a crosslinked Kv1.3).

Addition of data for ShK and Fab-ShK dissociation (slow, but similar) meets the request and helps validate use of the Fab-ShK, as well as Fig. S4

Good edit to emphasize no contacts with Kv1.3 extracellular loops.

Minor Comments: These are all appropriately amended.

REVIEWERS' COMMENTS

Reviewer #1 (Remarks to the Author):

The authors have addressed all of my previous concerns and clarified all questions I had. I am therefore simply restating what I previously said:

The manuscript by Selvakumar et al reports the structure of the voltage-gated potassium channel Kv1.3 in two, presumably, inactivated states at 2.89 Å, two states binding a Kv1.3 specific nanobody on the turret loops and a state blocked by a ShK-Fab construct. These are very important findings for the field because 1) Kv1.3 constitutes an attractive target for immunosuppression and the structures can enable the design of novel immunosuppressants and 2) the structures show novel, previously not captured selectivity filter conformations for Kv channels and thus provide novel insights into pore dynamics and the mechanisms of gating and permeation.

The manuscript is well written, the experiments beautifully conducted and the methods described in sufficient detail. The overall findings are very much in line with the biophysical properties of Kv1.3 and the ShK contacts are in agreement with previous findings. It, however, is a surprise how flexible the selectivity filter is and how many different, dilated, inactivated conformations exist. That is a tremendous new insight into the "life of Kv channels."

Reviewer #2 (Remarks to the Author):

The authors have substantially improved the manuscript. If the authors can address my two remaining comments, the manuscript will be suitable for publication in Nature Communications.

Line 270. The authors claim that Fab-ShK binding induces an active conformation of the selectivity filter. Is it also possible that the channel can sample the active conformation, even if transiently, and that binding changes the equilibrium between the active and distorted configurations? The authors should temper their conclusion in the text for the title for figure 5.

This is a good point and we agree. We have added text to communicate this point (Lines 26 and 277) and tempered the title for Figure 5.

Line 102. It would be better to avoid using an explicit resolution range for the reconstruction. While the plot suggests a maximum resolution of 2 Å, there are no features in the map consistent with a 2 Å reconstruction. Instead, state that the core of the protein is more well resolved compared to the periphery and especially to the T1 domains. Similar changes should be made when describing the local resolution of the other maps.

We agree and have updated the text accordingly. The changes can be found on Lines 103, 196, and a deletion was made on line 249.

Reviewer #3 (Remarks to the Author):

The authors have made substantial revisions to the manuscript, which should now be accepted. Brief comments are as follows.

1). General Comments:

a) The authors have addressed the issue of 150mM vs 2 mM K⁺ and their known effects on C-type inactivation. These are presented in Fig. S2. “Although these data show increasing concentrations of external K⁺ slow the entry...and diminishes the fraction of inactivated channels at 0 mV, most channels remain inactivated at 100mM external K⁺ and 0 mV...” Yes, these added conditions are similar to those used for the cryo-EM determinations. This allows them to say “D2 conformation... is likely to represent an inactivate state...” Further justification is given by alluding to structure similarities to other published inactivated Kv channels. The statement that D1 “may be an intermediate between the inactivated state and the conducting state” is speculation, but all right for now.

2). Addressed.

3). The question regarding use of C4 model when the majority of tetramers are a mixture of D1 and D2 subunits (D1:D2 1:3, 30.8%; D1:D2 2:2, 24.5%) was addressed, although not wholly satisfactorily. As the authors point out, no single tetramer model is able to faithfully represent the conformational heterogeneity in the structure, and they offer a practical way to present the results, fully acknowledging the problem in lines 126-130. Good answer to symmetry constraints (lines 189-192 & C1 symmetry map addition to Fig. S3A).

4). Good response to issue of D2 semblance to mutant Shaker and chimera with addition of Fig. S7 as well as the newly added electrophysiological data at high K⁺.

5). All the new labels and distances added to Figs. 2 & 3 are enormously helpful and allow specific comparisons. Additions to Figs 2G-I, 3J, 5H, and the new Fig. S7 address the request for a more rigorous comparison of pores. These details give a more substantial and quantitative description of the structural differences between chimera, D1, and D2, which ultimately is necessary for further mechanistic interpretations going forward. Likewise, the detailed comparison D1 and D3 is informative.

It would be good if some of the discussion in the rebuttal could be included in the Ms, but I understand if the authors assume the reader can infer these conclusions from the added labels/inserted distances and prefer not to include a summarizing paragraph.

Indeed, we agree with the reviewer that readers will be able to infer the conclusions from the labels and distance measurements in the figures.

6). Excellent response. Author texts on Lines 90-94 and Lines 240-243 clarify the rationale for engineering/using Fab-ShK fusion and the technical reasons (obviates obstacles to cryo-EM image processing due to properties of a crosslinked Kv1.3).

Addition of data for ShK and Fab-ShK dissociation (slow, but similar) meets the request and helps validate use of the Fab-ShK, as well as Fig. S4

Good edit to emphasize no contacts with Kv1.3 extracellular loops.

Minor Comments: These are all appropriately amended.